# Omni-iEEG: A Large-Scale, Comprehensive iEEG Dataset and Benchmark for Epilepsy Research

**Chenda Duan**[1][*] **Yipeng Zhang**[1][*] **Sotaro Kanai**[2] **Yuanyi Ding**[1] **Atsuro Daida**[2]
**Pengyue Yu**[1] **Tiancheng Zheng**[1] **Naoto Kuroda**[3] **Shaun A. Hussain**[2] **Eishi Asano**[3]
**Hiroki Nariai**[2] **Vwani Roychowdhury**[1][†]

[1]UCLA Samueli School of Engineering
[2]UCLA Mattel Children's Hospital, David Geffen School of Medicine
[3]Children's Hospital of Michigan, Wayne State University School of Medicine

## Abstract

Epilepsy affects over 50 million people worldwide, and one-third of patients suffer drug-resistant seizures where surgery offers the best chance of seizure freedom. Accurate localization of the epileptogenic zone (EZ) relies on intracranial EEG (iEEG). Clinical workflows, however, remain constrained by labor-intensive manual review. At the same time, existing data-driven approaches are typically developed on single-center datasets that are inconsistent in format and metadata, lack standardized benchmarks, and rarely release pathological event annotations, creating barriers to reproducibility, cross-center validation, and clinical relevance. With extensive efforts to reconcile heterogeneous iEEG formats, metadata, and recordings across publicly available sources, we present **Omni-iEEG**, a large-scale, presurgical iEEG resource comprising **302 patients** and **178 hours** of high-resolution recordings. The dataset includes harmonized clinical metadata such as seizure onset zones, resections, and surgical outcomes, all validated by board-certified epileptologists. In addition, Omni-iEEG provides over 36K expert-validated annotations of pathological events, enabling robust biomarker studies. Omni-iEEG serves as a bridge between machine learning and epilepsy research. It defines clinically meaningful tasks with unified evaluation metrics grounded in clinical priors, enabling systematic evaluation of models in clinically relevant settings. Beyond benchmarking, we demonstrate the potential of end-to-end modeling on long iEEG segments and highlight the transferability of representations pretrained on non-neurophysiological domains. Together, these contributions establish Omni-iEEG as a foundation for reproducible, generalizable, and clinically translatable epilepsy research. The project page with dataset and code links is available at `https://omni-ieeg.github.io/omni-ieeg/`.

## 1 Introduction

Epilepsy affects approximately 3.4 million people in the U.S. and nearly 50 million globally, making it one of the most common neurological disorders (for Disease Control & Prevention, 2017; Organization, 2023). About 30% of patients have drug-resistant epilepsy, where seizures cannot be controlled by medication (Kwan et al., 2010). Most of these patients experience focal seizures originating in specific brain regions (Jobst & Cascino, 2018), and successful treatment relies on accurately identifying the epileptogenic zone (EZ), the brain area crucial for seizure generation. There are two primary interventions aimed at disrupting or removing the EZ: (i) implantation of electrodes for targeted electrical stimulation and (ii) surgical resection of the affected brain tissue (Clinic, 2024). However, both strategies carry significant risks, including cognitive deficits resulting from damage to functionally critical regions (e.g., eloquent cortex) (Helmstaedter & Elger, 2013). Localization of the

---

[*]Equal contribution.
[†]Corresponding author: `<vwani@ucla.edu>`.

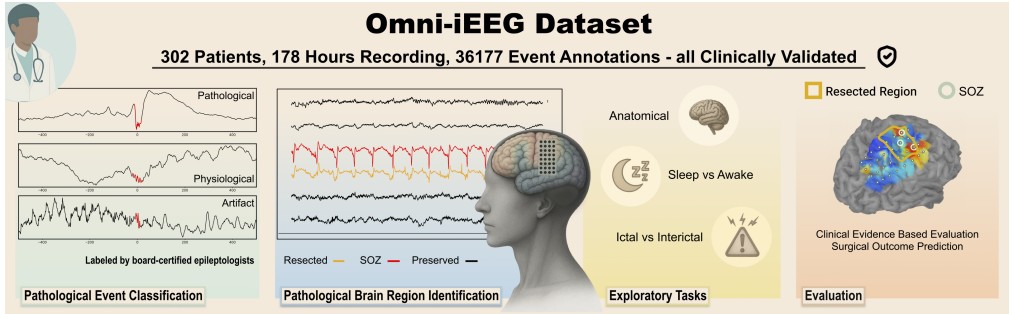

Figure 1: Overview of the Omni-iEEG Dataset and Benchmark.

EZ is typically guided by a combination of inpatient observation, neuroimaging, and intracranial EEG (iEEG), including identification of seizure onset zones (SOZ) and interictal spikes. Yet, SOZ-based resections do not guarantee seizure freedom (Rosenow & Lüders, 2001), and non-invasive tests, such as scalp EEG, MRI, PET, and MEG, often fail to localize the EZ with sufficient precision (Jayakar et al., 2016). The current clinical standard involves iEEG studies to identify both the pathological brain regions (i.e., the EZ) and the functional anatomical areas that must be preserved to minimize cognitive side effects. However, this process relies heavily on manual review of extended iEEG recordings, which is time-consuming and subject to low inter-rater reliability (Spring et al., 2017).

Several recent studies explore the use of machine learning on iEEG data or machine learning refined neurophysiological biomarkers to facilitate epilepsy research; for example, network analysis (Partamian et al., 2025) and Convolutional Neural Network (Li et al., 2021b; Zhang et al., 2022b). However, many of these efforts have been validated only on single-institution datasets with limited cohort sizes, restricting their clinical generalizability and robustness. Although datasets from institutions such as (Fedele et al., 2017; Zhang et al., 2025a; Bernabei et al., 2023a; Gunnarsdottir et al., 2022) have been released, they differ in data formats and inconsistent channel naming and demographic metadata. Moreover, benchmark and evaluation metrics are not standardized across studies, limiting reproducibility and comparability. These inconsistencies hinder the ability to derive translatable insights and to establish reliable benchmarks for model performance across studies.

To address these challenges, we construct **Omni-iEEG**, a large-scale, standardized dataset for epilepsy research. Omni-iEEG comprises recordings from **302 patients** across **178 hours** from eight leading epilepsy centers, including the University of California, Los Angeles; Wayne State University; the University Hospital Zurich; the University of Pennsylvania; the University of Miami; the National Institutes of Health; and Johns Hopkins Hospital. All recordings were obtained *prior to surgical resection* from patients with focal epilepsy, enabling models to predict postsurgical outcomes from pre-operative data to simulate the surgical planning. Furthermore, since the iEEG recording comes with different formats and metadata, board-certified epileptologists (experts) verified and harmonized all recordings with consistent iEEG formats, channel annotations, and clinical metadata. All data were fully de-identified with institutional IRB approval or public-domain release agreements.

Beyond the recordings and metadata, Omni-iEEG also releases clinically meaningful pathological biomarkers, extensively annotated by board-certified experts to support robust biomarker research. We focus on one of the most promising clinically utilized iEEG biomarkers for localizing the epileptogenic zone: high-frequency oscillations (HFOs). HFOs have garnered growing interest in both clinical (Gotman, 2010; Zweiphenning et al., 2022; Frauscher et al., 2018b) and computational (Kuroda et al., 2021; Sciaraffa et al., 2020; Chaibi et al., 2014; Daida et al., 2025) domain. Despite their potential, the clinical utility of HFOs remains contested due to persistent challenges in distinguishing pathological from physiological HFOs (Zijlmans et al., 2012; Zweiphenning et al., 2022), as well as issues such as artifact contamination and inter-rater variability (Nariai et al., 2018; Spring et al., 2017; Zhang et al., 2025b). Specifically, Omni-iEEG releases annotations of candidate HFOs, focusing on the most widely accepted pathological definition: HFOs co-occurring with spikes (spkHFO). These annotations are conducted on machine-generated detections from multiple widely used HFO detection algorithms (Navarrete et al., 2016; Ding et al., 2025).

To promote reproducible research and systematic model evaluation, we define clinically meaningful benchmark tasks with corresponding evaluation metrics. The two primary tasks are: (1) *Clinical Prior-Driven Pathological Event Classification*, which focuses on identifying iEEG HFO events consistent with expert-defined pathological patterns; (2) *Pathological Brain Region Identification*, which aims to localize regions likely involved in seizure generation. In addition, we propose a set of exploratory tasks, including *Anatomical Location Classification*, *Ictal Period Identification*, and *Sleep–Awake Classification* that hold comparable clinical significance yet present greater analytical challenges for iEEG.

We rigorously build baseline models from both machine learning and computational neuroscience domains, leveraging learning from clinically validated biomarkers as well as direct end-to-end data-driven approaches. Omni-iEEG serves as the first comprehensive dataset bridging epilepsy research and machine learning, with the potential to accelerate clinically relevant discoveries and improve patient outcomes. Our contributions can be summarized as follows:

- We introduce the Omni-iEEG dataset, a large-scale, standardized collection of interictal pre-surgical iEEG recordings from 302 patients and 178 hours of data, accompanied by rigorous clinical metadata verification, and includes over 36K expert annotations of pathological events.

- We define comprehensive evaluation metrics grounded in clinical priors and compare baseline models from diverse disciplines, providing a unified benchmark that improves reproducibility, enables comparability across studies, and bridges clinical research with machine learning.

- We demonstrate new insights enabled by Omni-iEEG, showing that end-to-end data-driven modeling of long iEEG segments can match or surpass clinically grounded biomarkers in epilepsy surgical planning, and reveal cross-domain transfer from pretrained audio representations to iEEG.

## 2 RELATED WORK

**Public iEEG Datasets.** Several public iEEG datasets, such as Open iEEG (Zhang et al., 2025a), Zurich iEEG HFO (Fedele et al., 2017), HUP (Bernabei et al., 2023a) and Epilepsy Interictal (Gunnarsdottir et al., 2022), provide valuable resources for epilepsy research. However, these datasets suffer from inconsistencies in metadata annotation and formats, making them challenging for machine learning applications without clinical expertise. There are more open-sourced iEEG data available (Li et al., 2021a; Huang et al., 2024; Berezutskaya et al., 2022; Wang et al., 2024; Madan et al., 2024), but out of the scope of epilepsy research, or containing only ictal recordings. The Omni-iEEG dataset addresses these issues by offering harmonized data across multiple centers, improving accessibility and usability for both clinicians and researchers.

**Clinically Discovered iEEG Biomarkers.** HFOs are well-established biomarkers for localizing epileptogenic zones, though distinguishing pathological from physiological HFOs remains challenging due to artifact contamination and inter-rater variability (Gotman, 2010; Frauscher et al., 2018b; Zijlmans et al., 2012; Nariai et al., 2018). Despite these challenges, HFOs remain central to iEEG-based epilepsy localization (Navarrete et al., 2016). Other interictal biomarkers also contribute to localizing epileptogenic zones, including interictal spikes and sharp waves (Doose et al., 1996), infraslow activity (Rodin et al., 2014), interictal epileptiform discharges (de Curtis et al., 2012), and phase-amplitude coupling (Samiee et al., 2018); however, these have seen limited adoption in recent clinical research. In the Omni-iEEG dataset, we focus on HFOs, as they are prevalent in existing literature and studies, enabling better integration of clinical findings with machine learning approaches for future research.

**Machine Learning-Based Data-Driven iEEG Biomarkers.** Machine learning techniques, including deep learning models, have shown promise in classifying iEEG signals and identifying epileptogenic regions (Li et al., 2021b; Monsoor et al., 2023; Partamian et al., 2025; Sheikh et al., 2024; Wang et al., 2022). However, these models often rely on small, single-institution datasets, limiting generalizability. The Omni-iEEG dataset, with its multi-center data, addresses these limitations and supports the development of more robust, clinically applicable models.

# 3 OMNI-iEEG DATASET

The Omni-iEEG dataset aggregates recordings from multiple independently curated datasets, spanning a diverse set of institutions and recording devices. The datasets vary in sampling rates, recording durations, modalities, and the availability of clinical metadata, such as seizure onset zones, resection zones, and surgical outcomes. A summary of the dataset statistics is provided in Table 1.

Table 1: Overview of the Omni-iEEG dataset. "#Subjects" indicates the total number of subjects in the dataset; "#Selected" denotes those included in the current split; "#Surgery" refers to subjects who underwent resective surgery; "#Seizure-Free" indicates subjects achieves post-operative seizue-free; "Duration" specifies the range of recording lengths in minutes.

| Split | Dataset | #Subjects | #Selected | #Surgery | #Seizure-Free | Duration (Min) |
|-------|---------|-----------|-----------|----------|---------------|----------------|
| Training | OpeniEEG | 111 | 111 | 97 | 66 | 5-111 |
|  | Zurich | 12 | 12 | 9 | 9 | 1-8 |
|  | HUP | 42 | 9 | 9 | 6 | 5 |
|  | SourceSink | 21 | 20 | 12 | 10 | 1-16 |
| Testing | OpeniEEG | 74 | 74 | 67 | 48 | 5-120 |
|  | Zurich | 8 | 8 | 6 | 6 | 3-12 |
|  | HUP | 16 | 6 | 6 | 4 | 5 |
|  | SourceSink | 18 | 15 | 9 | 6 | 1-17 |
| **Total** |  | **302** | **255** | **215** | **155** | — |

**Patient Cohort.** Since current open-source iEEG datasets are independently collected and released by different institutions, we conduct a comprehensive review of recently published epilepsy studies and assembled a unified collection of iEEG recordings to compose our *Omni-iEEG* dataset. Each dataset source is carefully cross-referenced with its corresponding publication to ensure consistency between the reported data specification and the shared data. We select high-quality recordings from each source by prioritizing those with reliable metadata alignment, particularly clinical channel annotations. The patient cohort included in the Omni-iEEG dataset is primarily drawn from the following data sources: Open iEEG Dataset (Zhang et al., 2025a), Zurich iEEG HFO Dataset (Fedele et al., 2017), Epilepsy Interictal Dataset (Gunnarsdottir et al., 2022), HUP dataset (Bernabei et al., 2023a). After our rigorous cleaning and validation, we aggregate a total of **302** patients and **178** hours of iEEG recordings. The detailed specification of the patient cohort is presented in the Appendix I.

**Data Processing and Harmonization.** We adopt the standardized preprocessing protocols defined by each source dataset when constructing the Omni-iEEG collection (Fedele et al., 2017; Gunnarsdottir et al., 2022; Zhang et al., 2025a; Bernabei et al., 2023a). All recordings are reviewed by board-certified clinicians to ensure data quality. To address heterogeneity across datasets, clinicians harmonize metadata into a unified schema, enabling consistent use. These data harmonization decisions are grounded in established clinical conventions and expert judgment: steps that are straightforward for clinicians but non-trivial for researchers without specialized training. All recordings in the released dataset are preserved at their original sampling rates as provided by the source datasets. For benchmarking and model training, we resample signals to 1000 Hz to standardize inputs; we provide per-recording sampling-rate metadata and a reference resampling script to reproduce this step consistently in downstream pipelines. Further details of validation, discrepancy resolution, and metadata harmonization are provided in the Appendix I; and an ablation study on the sampling rate is conducted in Appendix F.

**Data Splitting.** To facilitate standardized evaluation and ensure clinical relevance, we split the Omni-iEEG dataset into training and testing subsets using a 60%/40% ratio at the subject level. We carefully consider key metadata attributes to guide the split. Specifically, we ensure that patients from each contributing dataset are evenly distributed across the training and testing sets. In addition, we stratify the split to balance clinically meaningful variables, including resection outcomes, number of recording channels, and recording modalities to ensure that both subsets are representative of the overall cohort. For task-specific evaluations, the training and testing sets may be further restricted to subsets of the full splits to retain the original split logic wherever feasible.

Table 2: Summary of HFO event annotations. "#Subjects" denotes the number of patients; "#Total Events" refers to all annotated HFO events; "#Artifact" indicates events labeled as artifacts; "#Real HFO" represents real HFO; and "#spkHFO" refers to real HFOs that co-occur with spikes.

| Split | Dataset | #Subjects | #Total Events | #Artifact | #Real HFO | #spkHFO |
|---|---|---|---|---|---|---|
| Training | OpeniEEG | 20 | 19909 | 5517 | 14392 | 10070 |
| | Zurich | 3 | 829 | 242 | 587 | 449 |
| | HUP | 3 | 178 | 42 | 136 | 120 |
| | SourceSink | 3 | 1120 | 949 | 171 | 121 |
| Testing | OpeniEEG | 14 | 13092 | 2052 | 11040 | 7964 |
| | Zurich | 2 | 443 | 248 | 195 | 102 |
| | HUP | 2 | 91 | 82 | 9 | 4 |
| | SourceSink | 2 | 515 | 156 | 359 | 348 |
| **Total** | | **49** | **36177** | **9288** | **26889** | **19180** |

## 4 BENCHMARK TASKS FOR INTERICTAL iEEG ANALYSIS

### 4.1 TASK 1: CLINICAL PRIOR-DRIVEN PATHOLOGICAL EVENTS CLASSIFICATION

HFOs have been widely recognized as a key biomarker for localizing the epileptogenic zone in iEEG recordings. They are spontaneous iEEG events in the 80–500 Hz frequency range, characterized by at least four consecutive oscillations that clearly stand out from the background activity. HFO events vary in duration, typically ranging from 30 to 100 milliseconds (Zelmann et al., 2012). However, refining the clinical utility of HFOs requires further filtering of detected events, as legacy detection algorithms often produce noisy outputs (Zweiphenning et al., 2022). Due to the lack of consensus on the precise definition of HFOs, we adopt a consensus-driven approach by incorporating multiple mainstream detection algorithms (Navarrete et al., 2016) to create a comprehensive pool of candidate HFO events from multi-center iEEG recordings. Specifically, we utilize the Short-Time Energy detector (Staba et al., 2002), the Montreal Neurological Institute and Hospital detector (Zelmann et al., 2010), and the Hilbert HFO detector (Crépon et al., 2010) to maximize detection coverage.

Four board-certified clinicians manually annotate each candidate HFO event as either an artifact (including ringing, muscle, and background fluctuations), a pathological HFO, or a non-pathological HFO. We adopt the clinically promising definition of spkHFOs (Bénar et al., 2010), which are HFO events occurring in close temporal proximity to interictal epileptiform spikes. Although there is no consensus on the computational definition of spkHFOs (Zhang et al., 2022b), we use expert annotations from multiple clinicians across more than 36K candidate HFOs to develop a data-driven definition. Based on these annotations, the objective of this task is to perform multi-class classification to predict whether a given HFO event is a spkHFO, non-spkHFO, or artifact. To mitigate the effect of class imbalance across the three categories, we adopt macro-averaged precision, recall, F1 score, and area under the ROC curve (macro-AUC). The macro-AUC is computed in a one-vs-rest fashion and averaged equally across all classes. The classification dataset comprises **36,177** annotated HFO events, including **9,288** artifacts, **7,709** non-spkHFOs, and **19,180** spkHFOs. A summary of the annotation distribution is provided in Table 2, with additional details on the annotation protocol inter-rater agreement, and detector-specific statistics available in the Appendix B.

### 4.2 TASK 2: PATHOLOGICAL BRAIN REGION IDENTIFICATION

#### 4.2.1 MOTIVATION AND CLINICAL BACKGROUND

Accurate localization of pathological brain regions is critical for guiding effective epilepsy treatments, including surgical resection (Baumgartner et al., 2019) and neurostimulation (Johnson et al., 2022). Current clinical workflows, relying on prolonged iEEG monitoring, cortical stimulation, ictal recordings, and interictal biomarkers, are time-consuming, resource-intensive, and require extended inpatient monitoring and close coordination among clinical teams (Jayakar et al., 2016; Nariai et al., 2018).While HFOs are promising interictal markers of epileptogenicity, relying solely on them may overlook other informative iEEG features (Gunnarsdottir et al., 2022). A comprehensive data-driven approach that integrates spectral, temporal, and spatial characteristics has the potential to uncover

Table 3: Summary of channel labels and clinical outcomes across dataset splits. "#Subjects" indicates the number of valid subjects; "#Normal/Pathological Channels" represent iEEG channels labeled as normal or pathological; "#Seizure-Free/Non-Free" correspond to subjects with or without post-surgical seizure freedom; "#Anatomical Labeled" lists subjects with anatomical channel annotations.

| Split | Dataset | # Valid Subjects | # Normal Channels | # Pathological Channels | # Seizure Free | # Seizure Non-Free | # Anatomical |
|---|---|---|---|---|---|---|---|
| Training | OpeniEEG | 110 | 4638 | 1072 | 66 | 32 | 111 |
| | Zurich | 12 | 5619 | — | 9 | 3 | — |
| | HUP | 9 | 1034 | 106 | 6 | 3 | — |
| | SourceSink | 19 | 500 | 177 | 10 | 9 | — |
| Testing | OpeniEEG | 74 | 3533 | 575 | 48 | 19 | 74 |
| | Zurich | 8 | 2611 | — | 6 | 2 | — |
| | HUP | 6 | 874 | 78 | 4 | 2 | — |
| | SourceSink | 14 | 279 | 154 | 5 | 9 | — |
| | **Total** | **252** | **19088** | **2162** | **154** | **79** | **185** |

novel biomarkers and improve robustness, interpretability, and generalizability. Recent work (Gunnarsdottir et al., 2022; Wang et al., 2022; Chen et al., 2022) suggests that epileptogenic regions can be inferred from short interictal recordings. Although clinical decisions on resection or stimulation must still integrate multiple factors (Tamilia et al., 2018), automated analysis of interictal data offers a path toward streamlining pre-surgical evaluation and enhancing treatment planning.

### 4.2.2 TASK SPECIFICATION AND EVALUATION METRICS

**Clinical Evidence.** Since the precise localization of pathological brain regions cannot be determined prospectively (prior to surgery), the community often relies on established clinical evidence made retrospectively (after surgery) to guide the construction of this task (Zhang et al., 2022b): (1) SOZ channels, identified during monitoring, are expected to show interictal pathological activity; (2) Resected but non-SOZ channels are ambiguous, since margins depend on anatomical and surgical factors; (3) In seizure-free patients, preserved channels are presumed normal, as resection likely covered all pathological regions; (4) In non–seizure-free patients, residual pathological regions are inferred, though their exact locations remain unknown.

**Task Definition.** The objective of this task is to develop a biomarker or model that classifies iEEG channels as either pathological or normal. We introduce two evaluation criteria for this task: (1) the model should distinguish between channels with high-confidence labels: such as SOZ versus preserved channels in seizure-free patients; (2) it should provide patient-level predictions that are predictive of post-operative outcomes. Together, these evaluations aim to assess both the discriminative power of the model at the channel level and its clinical relevance.

**Evaluation at the Channel Level.** The channel classification label is defined as follows: Channels marked as SOZ during clinical monitoring are treated as positive examples, while preserved (non-resected) channels from seizure-free patients serve as negative examples. We report standard metrics including macro-averaged precision, recall (sensitivity), specificity (true negative rate), and the area under the ROC curve (AUC). Besides the AUC, given the clinical implications, we also place particular emphasis on both recall and specificity to minimize the risks associated with false negatives (missed pathological tissue) and false positives (unnecessary resection of healthy tissue).

**Surgical Outcome Prediction.** To evaluate whether the model's predictions correlate with surgical outcomes, following prior literature (Zhang et al., 2022a;b; 2025a; Gunnarsdottir et al., 2022; Monsoor et al., 2023), we compute a *resection ratio* (RR) for each patient. For every channel $c$, the model produces a pathological score $s_c$, and RR is defined as: $RR = \sum_{c \in \text{resected}} s_c / \sum_{c \in \text{all}} s_c$. This metric represents the proportion of predicted pathological signal that is surgically removed. We assign a binary outcome label to each patient: 1 for seizure-free (Engel Class I) and 0 for not seizure-free (Engel Class II–IV). We assess the predictive ability of the resection ratio using ROC-AUC, under the hypothesis that a higher RR corresponds to a higher likelihood of post-operative seizure freedom.

**Cohort Summary.** The pathological brain region identification task encompasses a total of **252** patients and **21250** candidate channels, comprising **2162** pathological (SOZ) and **19088** normal channels. Within these patients, **233** patients underwent resection treatment, and **154** of them became seizure-free after resection. A comprehensive summary of this task is presented in Table 3.

## 4.3 EXPLORATORY TASKS

Beyond the core benchmarks, we introduce three exploratory tasks that highlight clinically meaningful yet technically challenging problems in iEEG analysis. These tasks, though limited to data from one or two institutions, encourage new directions for data-driven epilepsy research. Full details and baseline results are in the Appendix C and D.

**Anatomical Location Classification.** This task aims to predict the anatomical location of iEEG signals using short segments. Accurate anatomical decoding can aid in functional mapping and surgical planning. Traditional methods rely on manual electrode labeling through stimulation, which is time-consuming and prone to variability. Automated classification improves precision and efficiency, overcoming these limitations (Bernabei et al., 2023b).

**Ictal Period Identification.** This task distinguishes between interictal and ictal periods in iEEG segments, which is crucial for localizing seizure onset zones and evaluating interventions. Manual identification is labor-intensive and prone to missing brief seizures, whereas automated detection improves diagnostic accuracy and timeliness (Fisher et al., 2014; Alhilani et al., 2020).

**Sleep-Awake Classification.** This task classifies iEEG segments into sleep or awake states, as seizure occurrence and interictal discharges are influenced by the sleep-wake cycle. Automated classification enhances diagnostic accuracy and treatment planning by overcoming the subjectivity and oversight inherent in manual classification (Derry & Duncan, 2013).

## 5 BENCHMARK RESULTS

### 5.1 PATHOLOGICAL EVENTS CLASSIFICATION

**Baseline Methods.** HFO events are high-frequency 1D signals with complex temporal and spectral dynamics. Previous interdisciplinary studies in HFO analysis (Zhang et al., 2024) have primarily focused on the morphology of iEEG traces, often using time-frequency representations (Morlet Wavelet). Building on this work (Zhang et al., 2024), we retrain their model on Omni-iEEG annotations, PyHFO-Omni. Additionally, we evaluate raw iEEG traces as a time-domain representation of HFO events, implementing baselines inspired by recent time-series models, including an LSTM+Attention (inspired by (Huang et al., 2023)), a Transformer based on the PatchTST architecture (Nie et al., 2023). We also train the competitive time-series framework, TimesNet (Wu et al., 2023a) on HFO event classification. Implementation details of each baseline model are provided in the Appendix E.

**Performance Comparison.** As shown in Table 4, the clinically validated framework, PyHFO-Omni, achieves the highest F1 score and recall, making it a strong baseline. However, time-domain baselines exhibit suboptimal performance. We hypothesize that since the spike (sharp waveform) and HFO characteristics lie within the high-frequency band, the subtle dynamics of the iEEG trace may not be fully captured by directly consuming the raw iEEG data. Additionally, time-domain baselines may pose difficulty in accounting for the correlations between different frequency bands, further complicating the learning task.

Table 4: Benchmark for Pathological Event Classification. Evaluation metrics include macro-averaged precision, recall, F1 and AUC.

| Model | Precision | Recall | F1 | AUC |
|---|---|---|---|---|
| LSTM+Attention | 0.7352 | 0.7359 | 0.7338 | 0.9109 |
| PatchTST Transformer | 0.7757 | 0.7686 | 0.7726 | 0.9311 |
| TimesNet | 0.7589 | 0.7726 | 0.7652 | 0.9221 |
| PyHFO-Omni | 0.8025 | 0.8110 | 0.8061 | 0.9390 |

## 5.2 PATHOLOGICAL BRAIN REGION IDENTIFICATION

To identify pathological brain regions, we evaluate two classes of baseline models: (1) approaches based on clinically validated biomarkers, HFO events (event-based models), and (2) purely data-driven approaches that directly learn from the iEEG segments (segment-based models).

**Event-Based Models.** Motivated by the clinical hypothesis that pathological regions exhibit higher rates of pathological HFOs, we first detect candidate HFO events (see Sec. 4.1) on each channel. To identify pathological HFOs, we use three models: (1) an eHFO classifier (Monsoor et al., 2023), which defines pathological HFOs using weakly supervised learning with clinical evidence, and (2) PyHFO-based classifiers, including the built-in model (Zhang et al., 2024) and our extended PyHFO-Omni (Sec. 5.1), both used to predict spkHFOs as pathological HFOs. After generating predictions, we compute the per-channel pathological HFO rate by counting pathological events. To derive binary channel-level predictions, we compute the ROC curve and use Youden's J statistic to select the optimal threshold. With this threshold, we obtain binary predictions and report macro-averaged precision, recall, specificity, and F1 score. For outcome prediction, we compute the resection ratio (RR) using the raw pathological rates and evaluate AUC, as defined in Sec 4.2.2.

**Segment-Based Models.** These models operate directly on raw iEEG segments without relying on intermediate event detection. During the training, we adopted a stratified sample to uniformly select the iEEG segments from each class. The segment duration is set to one minute, providing a broader spectrum than that of HFOs (normally 30-100ms). For evaluation, we use the models to conduct inference on non-overlapping 1-minute segments from each channel in a sliding window manner to assign a probability for each segment. Then we compute a per-channel pathological score as the average of the pathological probability. Metrics reported in channel-level evaluations follow the same scheme as event-based models. Meanwhile, RR is also calculated in the same manner as the event-based model to standardize the outcome prediction.

To establish an insightful and comprehensive benchmark for the segment-based model, we examine applications across diverse machine learning domains. We implement SEEG-NET (Wang et al., 2022), a model motivated by clinical knowledge for classifying pathological activity. Moreover, given the long duration of the 1-minute iEEG signals, which are one-dimensional in nature, we explore the fine-tuning of state-of-the-art audio processing frameworks Contrastive Language-Audio Pretraining (CLAP) (Wu et al., 2023b), on iEEG data. Furthermore, inspired by event-based models, we implement a vision-style architecture, denoted as TimeConv-CNN, which processes 1-minute time–frequency plots of 1000 Hz iEEG signals. The model first applies 1D convolution along the time axis to capture long-range temporal dynamics, followed by CNN layers for joint temporal–spectral feature extraction. Implementation details are provided in the Appendix E.

**Performance Comparison.** Table 5 summarizes results for pathological channel classification and postoperative outcome prediction. The event-based model grounded in clinical priors (PyHFO-Omni) and the segment-based model (TimeConv-CNN) trained end-to-end achieve similar outcome AUCs (0.74 vs. 0.73). Crucially, TimeConv-CNN not only matches outcome prediction but also outperforms PyHFO-Omni in pathological channel identification, achieving the highest channel-level AUC (0.81). By contrast, SEEG-NET attains strong classification AUC but low specificity, limiting its clinical utility. These findings highlight the need for multi-metric evaluation beyond outcome AUC, while also underscoring the potential of end-to-end approaches to transform iEEG-based epilepsy research.

## 5.3 BEYOND BENCHMARKING: TRANSLATIONAL INSIGHTS FROM OMNI-IEEG

**Clinical Generalizability.** Publicly available event-based models trained on single-center datasets perform poorly in our benchmark, underscoring the importance of multi-institutional datasets like Omni-iEEG for building clinically reliable and generalizable models.

**Direct Modeling of Minute-Long iEEG Segments.** Direct end-to-end modeling of 1-minute intracranial EEG recordings at kilohertz resolution has not been fully explored in prior work. Existing approaches have largely focused on short, event-centered windows (e.g., HFOs or spikes) or relied on heavily downsampled features, both of which discard long-range temporal dependencies. We address this challenge through two complementary strategies: (i) leveraging powerful representations from a different domain by treating iEEG as an audio signal, and (ii) designing a custom architecture, TimeConv-CNN, tailored to the unique scale and resolution of iEEG.

Table 5: Benchmark results for identifying pathological brain regions. Evaluation has two dimensions: (1) **Pathological Channel Identification**, based on clinical priors of pathological(SOZ channels) vs. normal (preserved channels in seizure-free patients); (2) **Outcome AUC**, which assesses whether higher scores in resected channels correlate with postoperative seizure freedom via the resection ratio.

| Model | Pathological Channel Classification | | | | | Outcome |
|---|---|---|---|---|---|---|
| | Precision | Recall | F1 | Specificity | AUC | AUC |
| **Event-Based Models** | | | | | | |
| eHFO | 0.6053 | 0.6466 | 0.6195 | 0.4101 | 0.6611 | 0.4521 |
| PyHFO$_{spkHFO}$ | 0.6000 | 0.6431 | 0.6140 | 0.4089 | 0.6557 | 0.4972 |
| PyHFO-Omni$_{spkHFO}$ | 0.5799 | 0.6991 | 0.5635 | 0.6951 | 0.7351 | 0.7438 |
| **Segment-Based Models** | | | | | | |
| SEEG-NET | 0.5790 | 0.7169 | 0.5259 | 0.6049 | 0.7850 | 0.5952 |
| CLAP | 0.5936 | 0.6997 | 0.6009 | 0.7823 | 0.7684 | 0.6770 |
| TimeConv-CNN | 0.6259 | 0.7454 | 0.6469 | 0.8230 | 0.8061 | 0.7380 |

**Cross-Domain Transfer with Audio-Pretrained Models.** Remarkably, CLAP, an audio-pretrained model optimized for generic acoustic signals rather than neurophysiology, achieves competitive performance on pathological channel classification after fine-tuning. This highlights the cross-domain generalization ability of audio representations to iEEG, where long-sequence dynamics can be repurposed as clinically meaningful features.

**TimeConv-CNN for Multi-Resolution Dynamics.** To capture both long-range temporal dependencies and fine-grained high-frequency characteristics, we introduce TimeConv-CNN. This architecture applies temporal convolutions directly to the large time–frequency representation (60,000 time points × 224 frequency components), enabling efficient compression while preserving salient dynamics across scales. With an efficient yet expressive design, TimeConv-CNN opens a new pathway for identifying pathological brain regions, where both coarse and detailed patterns are critical.

**A Potentially "Hear-able" Biomarker.** The competitive performance of fine-tuning an audio-pretrained model like CLAP prompted a more direct question: If iEEG signals can be processed like audio, could their pathological features also be directly "heard"? To illustrate the potential for novel, cross-domain biomarker discovery, we conduct an exploratory case study on a representative patient (`sub-openieegUCLA01`). We convert one-minute iEEG segments into audio waveforms and use an audio classifier (YAMNet) (Gemmeke et al., 2017) to label them. Specifically, each segment is normalized with z-scoring, time-compressed by a factor of 10, resampled to 16 kHz, and peak-normalized to produce standardized audio waveforms. The results are striking: among 3,128 one-minute segments, including 276 from SOZ channels and 2,852 from preserved channels; in the top-1 prediction, YAMNet labels 87 SOZ segments as "helicopter" while never assigning this label to preserved channels; expanding to the top-5 predictions, "helicopter" appears in 159 segments overall, compared to only 5 labeled as "physiological." While this N=1 observation may not be generalizable, it serves as a compelling proof-of-concept that clinically relevant neural dynamics may possess intuitive, "hearable" signatures. This opens a novel and highly interpretable direction for future biomarker discovery.

## 6 CONCLUSION AND LIMITATIONS

We present **Omni-iEEG**, a large-scale, pre-surgical iEEG dataset comprising 302 patients and 178 hours of high-resolution recordings. This dataset is thoroughly verified with clinical metadata with released annotations of the pathological HFO biomarker. We define two primary and three exploratory clinically meaningful benchmarks, grounded in clinical insights established in the literature. We further benchmark multiple baselines across various disciplines, leveraging both clinically defined biomarkers and end-to-end data-driven approaches for a comprehensive evaluation. Beyond quantitative performance, our benchmarking also yields translational insights, highlighting opportunities for novel biomarkers and cross-domain modeling strategies with direct clinical relevance.

Despite the significance and utility of the Omni-iEEG dataset and its associated benchmarks, several limitations remain. These include potential gaps in the clinical insights captured and the inherent subjectivity of spkHFO annotations, even when annotators reached consensus with high inter-rater agreement in this study. Additionally, while the benchmarks cover key tasks in epilepsy research, certain alternative HFO detection methods and artifact removal strategies were not explored.

These limitations largely reflect the breadth of epilepsy research, where decades of diverse methodologies cannot be encompassed in a single study. Nonetheless, they underscore the importance of Omni-iEEG in providing a standardized foundation for comprehensive evaluation and a platform for future exploration. We anticipate that our effort will not only support more reproducible benchmarking but also catalyze new methods, biomarkers, and clinical insights that advance both machine learning and epilepsy treatment.

Further details, including event annotation procedures, dataset harmonization and specifications, exploratory tasks, implementation details, interpretation analyses, ablation studies such as cross-dataset generalization, and an extended discussion of limitations, are provided in the Appendix.

ETHICS STATEMENT

The Omni-iEEG dataset is principally constructed from four publicly available resources hosted on OpenNeuro: the Open iEEG Dataset, the Zurich iEEG HFO Dataset, the Epilepsy Interictal Dataset, and the HUP dataset. All data were fully de-identified with institutional IRB approval or public-domain release agreements. Each of these datasets is released under a CC0 license, thereby permitting processing, integration, and redistribution, provided that the original sources are appropriately acknowledged.

REPRODUCIBILITY STATEMENT

The Omni-iEEG dataset has been reorganized in accordance with the Brain Imaging Data Structure (BIDS) standard and will be made publicly available via *Hugging Face*. To support reproducibility and adoption, we will also release an accompanying software library that provides utilities for streamlined data loading, flexible filtering using clinical metadata, and ready-to-use training and evaluation pipelines for benchmarking and model development. Furthermore, model checkpoints corresponding to all baseline methods will be made available to facilitate comparison and future research.

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

APPENDIX

CONTENTS OF APPENDIX

## A  THE USE OF LARGE LANGUAGE MODELS

We used Large Language Models, such as ChatGPT, as general-purpose assist tools to polish the writing (grammar, phrasing, and readability) and to aid in literature discovery by surfacing potentially relevant related work. All scientific ideas, experimental designs, analyses, and conclusions were developed by the authors, and LLMs did not contribute at the level of a scientific contributor or author. The authors take full responsibility for the entirety of the paper's content.

## B  PATHOLOGICAL EVENTS ANNOTATION

The annotation protocol used in the current study adopt a consistent and standardized protocol from the prior studies (Nariai et al., 2018) for event annotation. Regarding patient selection and detector choice, we introduce diversity to ensure broad coverage across varying conditions and recording setups.

**Subject Selection.** Patients are selected from diverse clinical scenarios in the Omni-iEEG dataset, including those undergoing different recording modalities and from multiple institutions. The overall annotations include iEEG recording from eight centers in Omni-iEEG dataset, with both ECoG and SEEG recordings represented. A summary of the patient and event distributions can be found in Table 4 of the main paper. Events detected by the Short-Term Energy (STE), Montreal Neurological Institute (MNI), and Hilbert transform–based algorithms were independently annotated to verify the consistency of the proposed methods across detection approaches. We adopted previously validated HFO detection parameters from published work (Zhang et al., 2025a). The complete parameter settings are summarized in Table 6.

Table 6: Detector parameters used in our annotation pipeline. All frequency units are in Hz, time in seconds unless noted.

| STE Detector | | MNI Detector | | Hilbert Detector | |
|---|---|---|---|---|---|
| Parameter | Value | Parameter | Value | Parameter | Value |
| filter_freq | [80, 300] | filter_freq | [80, 300] | filter_freq | [80, 300] |
| rms_window | $3 \times 10^{-3}$ | epo_CHF | 60 | sd_thres | 5 |
| min_window | $6 \times 10^{-3}$ | per_CHF (%) | 95 | min_window | $10 \times 10^{-3}$ |
| min_gap | $10 \times 10^{-3}$ | min_win | $10 \times 10^{-3}$ | epoch_len | 3600 |
| min_osc | 6 | min_gap | $10 \times 10^{-3}$ | | |
| rms_thres | 5 | thrd_perc | 99.9999% | | |
| peak_thres | 3 | base_seg | $125 \times 10^{-3}$ | | |
| epoch_len (ms) | 600 | base_shift | 0.5 | | |
| | | base_thrd | 0.67 | | |
| | | base_min | 5 | | |
| | | epoch_time (ms) | 10 | | |

**Annotation Procedure.** Annotations were conducted by four board-certified clinical epileptologists, each with extensive experience in epilepsy surgery and iEEG interpretation. All experts are actively involved in clinical epilepsy research, including patient diagnosis and surgical planning, ensuring that the annotations reflect high clinical standards and real-world applicability.

To standardize and streamline the workflow, all iEEG recordings were resampled to 1 kHz, ensuring a consistent temporal resolution that facilitates filtering, feature extraction, and cross-patient comparisons. Bipolar montage is applied to the Zurich, HUP, and Sourcesink recordings. Candidate HFO events are first detected using PyHFO, and each event is subsequently reviewed and labeled by the experts as either artifact, non-spkHFO, or spkHFO.

To ensure inter-rater reliability, the experts first collaboratively reviewed a shared subset of sample events to establish clear labeling criteria and achieve preliminary consensus. Following this calibration, each annotator independently labeled their assigned data subsets.

To further evaluate consistency, three primary annotators (R1–R3) independently labeled a shared validation set of 200 events. Inter-rater agreement was high, with a Fleiss' $\kappa = 0.92$ across the three annotators. Pairwise Cohen's $\kappa$ ranged from 0.88 to 0.93 (Table 7). A complementary assessment

between R4 and R3 on 583 events showed similar reliability (Cohen's $\kappa = 0.90$ for artifact vs. non-artifact; $\kappa = 0.82$ for spk vs. non-spk).

Table 7: Inter-rater agreement among the three primary annotators. Pairwise agreement is measured using Cohen's $\kappa$, and overall agreement is measured using Fleiss' $\kappa$.

| Annotator(s) | Metric | Value |
|---|---|---|
| R1 vs. R2 | Cohen's $\kappa$ | 0.9427 |
| R3 vs. R1 | Cohen's $\kappa$ | 0.9243 |
| R3 vs. R2 | Cohen's $\kappa$ | 0.9091 |
| R1, R2, R3 | Fleiss' $\kappa$ | 0.9254 |

## C  EXPLORATORY TASKS

### C.1  ANATOMICAL LOCATION IDENTIFICATION

#### C.1.1  MOTIVATION AND CLINICAL BACKGROUND

Accurate anatomical localization of iEEG electrodes is critical for both clinical decision-making and neuroscientific research. Traditionally, this process involves co-registering post-implantation CT scans with pre-implantation MRI images, followed by manual or semi-automated labeling using neuroimaging software tools (Blenkmann et al., 2017; Lucas et al., 2024). While effective, these methods are time-consuming, resource-intensive, and may not be feasible in all clinical environments due to the need for imaging infrastructure and expert interpretation. As a result, some publicly released iEEG datasets do not include electrode-level anatomical annotations, limiting their utility for downstream physiological and clinical analyses.

Anatomical localization plays a vital role in interpreting physiological patterns in iEEG data. Previous work (Guragain et al., 2018) has shown that certain electrophysiological events exhibit region-specific distributions: for example, physiological HFOs are predominantly observed in occipital regions, while sleep spindles typically originate from fronto-parietal areas (Andrillon et al., 2011). More critically, in the clinical context of epilepsy surgery, localizing electrodes relative to eloquent cortices, such as those supporting motor, language, or memory functions, is essential for preserving key functions during resection. Besides the aforementioned biomarkers located in different anatomical locations, multiple studies have demonstrated that rich interictal electrophysiological signatures are specific to anatomical brain regions. Cortical areas such as the hippocampus, frontal lobe, and occipital lobe exhibit characteristic spectral and complexity features that are reproducible across patients (Frauscher et al., 2018a; Groppe et al., 2013). These region-specific patterns are often clinically observable, and prior work has shown that even short segments, as little as five minutes, of interictal data can be sufficient to distinguish between regions (Frauscher et al., 2018a). Building on these insights, recent machine learning methods have demonstrated promising classification of anatomical regions using only signal-based features, showcasing the feasibility of non-imaging-based brain mapping (Taylor et al., 2022). Thus, the ability to infer anatomical locations from electrophysiological recordings holds significant potential not only for enhancing physiological interpretations but also for informing clinical decisions, particularly in datasets that lack explicit anatomical metadata. Motivated by these findings, we propose an anatomical location identification task to benchmark and encourage further development of non-imaging-based localization approaches.

#### C.1.2  TASK DEFINITION AND EVALUATION METRICS

**Task Definition.** Inspired by above findings, this task aims to predict the anatomical location of iEEG electrodes based on one minute of interictal signal per channel. We formulate this as a supervised classification problem at two levels of spatial granularity. The first level involves coarse-grained classification into **five major anatomical regions**: *frontal, temporal, parietal, limbic, and occipital lobes*. The second level refines this prediction to **twelve fine-grained subregions**, including *Cingulate, Motor/premotor, Occipital, Mesial temporal, Inferior temporal, Inferior parietal, Prefrontal, Somatosensory, Superior parietal, Superior temporal, Insula, and Other/Subcortical*.

**Evaluation Metrics.** Since anatomical localization is performed at the channel level, similar to the pathological localization task described in the main manuscript, we adopt a channel-wise aggregation strategy to evaluate model performance. Specifically, we average the model-predicted probabilities across all 1-minute windows for each channel, yielding an overall probability distribution over anatomical locations (major: 5 classes; fine: 12 classes) per channel. Performance is then assessed using standard classification metrics over both label granularities. We report balanced accuracy (Acc) and macro-averaged F1 score (F1), which are robust to class imbalance and reflect performance across all anatomical regions.

**Cohort Summary.** Among the datasets considered, only the Open-iEEG dataset includes anatomical labels for individual channels. Therefore, this downstream task is restricted to two research centers within the Open-iEEG dataset, following the patient-wise cross-validation split outlined in Section 3 of the main manuscript. Specifically, the anatomical localization task involves 185 patients and 18517 candidate channels, each with known anatomical labels obtained through imaging co-registration and expert annotation.

## C.2   Ictal Period Classification

### C.2.1   Motivation and Clinical Background

Accurate identification of ictal periods, moments when seizures are actively occurring, is fundamental to epilepsy diagnosis and treatment planning. Clinically, the transition from interictal to ictal states provides crucial insight into seizure dynamics, including onset, propagation, and termination (Wendling et al., 2005). However, ictal events can be brief, subtle, and easily overlooked in lengthy recordings (Pyrzowski et al., 2021). Manual review by experts is time-consuming and may miss atypical or subclinical seizures, which are increasingly recognized as clinically relevant (Fisher et al., 2014; Alhilani et al., 2020; Sumsky & Greenfield Jr, 2022; Kharbouch et al., 2011). An automated system that reliably distinguishes ictal from interictal activity offers not only diagnostic support and improved consistency across reviewers and institutions but also has the potential to reduce overall monitoring duration and alleviate the burden on clinical staff.

Moreover, recent advances in neuromodulation therapy, such as responsive neurostimulation (RNS), highlight the need for real-time, fine-grained seizure detection (Sun et al., 2008). Clinicians have noted that identifying precise ictal boundaries could help define personalized stimulation targets, enabling therapeutic interventions to occur at the earliest possible seizure onset (Boddeti et al., 2022). In this context, accurate ictal detection is not merely retrospective; it becomes a cornerstone for proactive, closed-loop treatment systems. Therefore, we view ictal period identification as a clinically actionable task, bridging seizure monitoring with intervention and offering a foundation for both diagnostic and therapeutic innovation.

### C.2.2   Task Definition and Evaluation Metrics

**Task Definition.** The goal of this task is to distinguish between interictal and ictal periods iEEG recordings. The ictal recordings are contained in the HUP dataset, but the interictal recordings could be retrieved from all patients from the Omni-iEEG dataset. For each patient, we segment the data into one-minute clips and formulate the problem as a binary classification task: given a one-minute iEEG segment from all channels, predict whether it originates from an ictal or interictal recording. This formulation ensures a standardized input length while reflecting realistic clinical scenarios where seizure identification must occur from streaming or windowed recordings.

**Evaluation Metrics.** Since our baseline models operate on 1D signals while the task is defined at the recording level (classifying whether a multi-channel iEEG recording corresponds to an ictal or interictal state), we adopt a straightforward aggregation strategy. Due to the varying number of channels across patients, which complicates direct cross-channel neural network training, we compute the average predicted probability across all channels for each 1-minute window in the test set. This aggregated probability is treated as the model's prediction for the ictal/interictal state of that window. We then determine the optimal classification threshold using the area under the ROC curve (AUC) and the Youden's J method. Using the binarized predictions from this threshold, we evaluate model performance using balanced accuracy (Acc) and macro-averaged F1 score (F1), which account

for class imbalance and provide a comprehensive assessment of performance across both ictal and interictal classes.

**Cohort Summary.** This task includes **16** patients in the training set and **9** patients in the test set. Notably, the interictal patients are uniformly sampled from each data source. The training set includes **6205** unique channels, while the test set comprises **3561** unique channels. To prevent overrepresentation from long recordings, we apply stratified sampling during training to draw windows evenly across channels.

### C.3 SLEEP AWAKE CLASSIFICATION

#### C.3.1 MOTIVATION AND CLINICAL BACKGROUND

Accurately identifying sleep and awake states in iEEG is essential for both clinical care and research. The vigilance state of a patient strongly influences the manifestation of seizures and interictal discharges; for example, frontal lobe seizures are more likely to occur during sleep, particularly in non-REM stages, whereas temporal lobe seizures are more frequent during wakefulness (Herman et al., 2001). In long-term recordings like SEEG, knowledge of the sleep–awake cycle can aid in distinguishing epileptic from non-epileptic events (Bazil & Walczak, 1997). From a clinical perspective, this information improves diagnostic precision and guides personalized treatment strategies, especially when manual annotations are inconsistent or missing (Moore et al., 2021). Moreover, sleep–awake identification is often more challenging in iEEG than in scalp EEG, where awake states are typically accompanied by distinct artifacts that assist visual classification (Lambert & Peter-Derex, 2023). In contrast, iEEG recordings lack such clear surface-level markers, making automated approaches particularly valuable (Lambert & Peter-Derex, 2023). Automated sleep staging enables efficient parsing of long iEEG recordings, reduces annotation burden, and helps identify longer usable segments for downstream analysis (Derry & Duncan, 2013).

Sleep–awake classification extends beyond clinical utility, enabling consistent, automated labeling of vigilance states across large iEEG datasets. This supports standardized analysis of sleep-dependent modulation of interictal biomarkers like high-frequency oscillations, facilitating cross-patient comparisons and advancing scalable, reproducible brain state analysis in epilepsy research (Zhao et al., 2024).

#### C.3.2 TASK DEFINITION AND EVALUATION METRICS

**Task Definition.** The goal of this task is to distinguish between sleep and awake periods in iEEG recordings. The awake recordings are in the Sourcesink dataset, but the sleep recordings can be retrieved from all patients in the Omni-iEEG dataset. For each patient, we segment the data into one-minute clips and formulate the problem as a binary classification task: given a one-minute iEEG segment, predict whether it originates from a sleep or awake recording. This formulation ensures a standardized input length and aligns with real-world clinical needs, where vigilance state identification must often be performed from streaming or windowed data.

**Evaluation Metrics.** Similar to the ictal/interictal classification task, the sleep/awake classification is also performed on multi-channel iEEG recordings. We adopt the same evaluation strategy, aggregating predictions across channels for each 1-minute window. Model performance is assessed using standard binary classification metrics, including balanced accuracy (Acc) and macro-averaged F1 score (F1), which provide a fair evaluation across both sleep and awake states.

**Cohort Summary.** This task includes **13** patients in the training set and **8** patients in the test set. The interictal patients are uniformly sampled from each data source. The training set includes **1751** unique channels, while the test set comprises **802** unique channels. We also apply stratified sampling during training to draw windows evenly across channels.

### D EXPLORATORY TASKS BASELINE RESULTS

The detailed benchmark results for all three tasks across baseline models, SEEG-NET, CLAP, and CNN, are presented in Table 8. Interestingly, the foundation audio embedding model CLAP often achieves strong performance across tasks, and the end-to-end CNN also demonstrates competitive

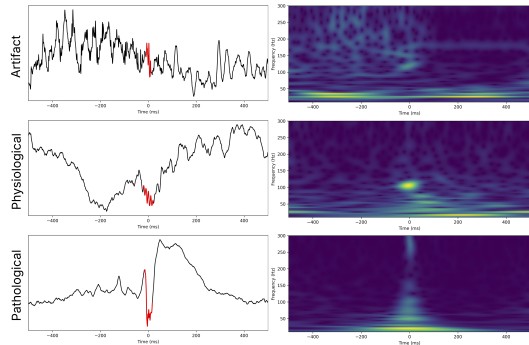

Figure 2: Event-based iEEG input: raw waveform (left) and time-frequency representation (right).

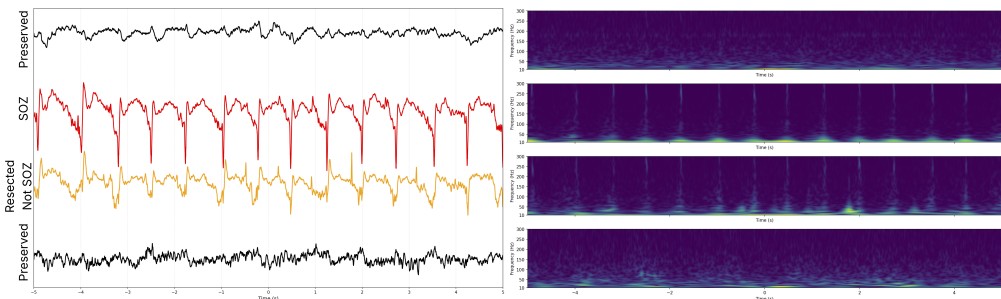

Figure 3: Channel Segment-based iEEG input: raw waveform (left) and time-frequency representation (right).

results, consistent with the trends observed in the main manuscript. We hypothesize that the generally better performance of these models reflects their stronger capacity to capture relevant signal representations. This suggests that the neurophysiological biomarkers underlying these tasks are embedded in the morphological characteristics of the iEEG signal, supporting the relevance of our proposed dataset and task formulation.

For SEEG-NET, since the authors did not release their implementation or dataset, we reproduced the architecture based on the details provided in the paper. However, the resulting performance was suboptimal, which may be attributed to potential discrepancies between our implementation and the original, given the absence of released code and data. This further highlights the importance of our benchmark, which offers a publicly available, reproducible platform for developing and evaluating models in this domain.

Table 8: Macro-averaged F1 and balanced accuracy scores for the Anatomical with 5 Classes and 12 Classes, Ictal/Interictal, and Sleep/Awake classification tasks.

| Model | Anatomical(5) | | Anatomical(12)) | | Ictal | | Sleep–Awake | |
|---|---|---|---|---|---|---|---|---|
| | F1 | Acc | F1 | Acc | F1 | Acc | F1 | Acc |
| SEEG-NET Wang et al. (2022) | 0.2520 | 0.2550 | 0.1081 | 0.1853 | 0.7526 | 0.7624 | 0.6773 | 0.6421 |
| CLAP Wu et al. (2023b) | 0.4750 | 0.4894 | 0.3540 | 0.3897 | 0.9245 | 0.9323 | 0.7225 | 0.9331 |
| TimeConv-CNN | 0.4788 | 0.4708 | 0.3087 | 0.3215 | 0.8533 | 0.8720 | 0.7118 | 0.9291 |

# E IMPLEMENTATION DETAILS

## E.1 TIME-FREQUENCY REPRESENTATION

Inspired by prior interdisciplinary studies (Zhang et al., 2022a; Ding et al., 2025), we adopt the time-frequency representation (spectrograms) of iEEG signals using Morlet wavelet transforms to

capture the temporal dynamics of each frequency band. This representation has been shown to provide a comprehensive view of signal morphology and demonstrated superior performance in both previous literature and our own experiments. As illustrated in the example time-frequency representation of HFO events (Figure 2) and different iEEG channels (Figure 3), distinct morphological features in the iEEG tracing become more distinguishable in the time-frequency domain. Moreover, the inherent parallelism of time-frequency computation enables efficient processing on GPUs, significantly accelerating feature extraction. Our implementation of this process is available in the accompanying code repository. For downstream tasks, we follow established practices from prior work (Zhang et al., 2024), which indicate that the most informative content in iEEG signals lies within the 10–300 Hz range when the downstream models take the time-frequency representation as inputs.

## E.2 Event-Based Models.

**Data Construction.** For each labeled HFO event, we extract a 570 ms iEEG segment centered at the event midpoint. To ensure consistency across recordings, we include only data with a minimum sampling rate of 1000 Hz and resample all inputs to 1000 Hz. These standardized segments serve as the input to all baseline models.

**PyHFO-Omni Architecture.** Following the architecture of the HFO classification model introduced in PyHFO (Zhang et al., 2024), we retrain the model using Omni-iEEG annotation. Specifically, we use a convolutional neural network based on a modified ResNet18 backbone for event-level classification. The input spectrograms are passed to a ResNet18 encoder, where the first convolutional layer is adjusted to accept single-channel input, and the classification layer produces the logit of each class.

**Attention+LSTM Architecture.** Inspired by Huang et al. (2023), we implement a bidirectional LSTM model for event-level classification on raw iEEG signals. The input sequence is first processed by a two-layer bidirectional LSTM to generate hidden states at each timestep. To summarize the temporal dynamics, we compute a weighted sum of the hidden states using learned additive weights. This aggregated context vector is then passed through classification layers to produce the final output logits.

**PatchTST Transformer Architecture.** We implement a PatchTST-style (Nie et al., 2023) Transformer model for raw iEEG signal classification. Each 1D input sequence is split into non-overlapping patches, which are linearly projected into a fixed embedding space. A learnable class token is prepended to the patch sequence, and positional encodings are added to retain temporal order. The resulting sequence is processed by a multi-layer Transformer encoder. The output embedding corresponding to the class token is then passed through a classification head to produce the final logits.

**TimesNet Architecture.** We include TimesNet (Wu et al., 2023a), a recent competitive time-series model that captures multi-periodic patterns using a frequency-aware design. The model first embeds the input sequence and passes it through multiple stacked modules, each identifying dominant periods via FFT and applying 2D convolutions on reshaped periodic representations. These outputs are adaptively aggregated based on learned frequency weights. We adopt the official implementation without modification and apply it directly to our event-level iEEG classification task.

For comprehensive details on model architecture and parameter configurations, please refer to our released codebase.

## E.3 Segment-Based Models.

**SEEG-Net Reproduction.** We implement SEEG-Net, a multiscale and attention-based deep learning architecture originally proposed for pathological activity detection in SEEG signals (Wang et al., 2022). The model begins with a multiscale convolutional block composed of three parallel 1D convolutional branches with varying kernel sizes, each capturing frequency-specific features at different temporal resolutions. Outputs from all branches are concatenated and passed through a residual squeeze-and-excitation (SE) block to enhance the most informative feature channels. This representation is then fed into a bidirectional LSTM with an attention mechanism to model long-range temporal dependencies. Finally, a feedforward classifier outputs a binary decision. Our

implementation follows the core architectural design of the original SEEG-Net and adapts it for our segment-level iEEG classification task.

**CLAP-Based Model.** To explore pretrained audio-language representations for iEEG classification, we adapt the CLAP model (Wu et al., 2023b), originally trained on natural audio-text pairs, for our long-form iEEG recordings. Specifically, we employ the pretrained *clap-htsat-fused* checkpoint and attach a lightweight classification head. Raw iEEG waveform segments are processed using the CLAP feature extractor with adjusted hyperparameters. We conduct full-parameter fine-tune of CLAP with a classification head. This setup leverages CLAP's pretrained acoustic representations for downstream iEEG classification tasks.

**TimeConv-CNN Architecture.** Long iEEG recordings yield extremely large time–frequency representations, which are rarely modeled directly due to their prohibitive scale. To address this challenge in a clinically meaningful setting, we design a TimeConv-CNN architecture that adapts established vision backbones to the unique demands of iEEG. The model begins with two *time-wise convolutional layers*, which apply vertical (temporal-only) convolutions across the spectrogram. This design specifically targets the elongated temporal axis, compressing long sequences while preserving detailed spectral structure that is critical for identifying pathological dynamics. The reduced representation is then consumed by a modified ResNet18 backbone, where the first convolutional layer is adapted to match the reshaped feature dimensions. This allows us to leverage the hierarchical feature extraction of ResNets without overwhelming computational resources. Finally, a classification layer produces the output logits. We adapt axis-specific convolutions as a practical strategy to model minute-long, kilohertz-sampled iEEG segments end-to-end. Temporal convolutions efficiently compress the elongated time axis, and the ResNet backbone captures higher-order spatiotemporal patterns, enabling scalable modeling of long iEEG recordings.

For comprehensive details on model architecture and parameter configurations, please refer to our released codebase.

### E.4    TRAINING HYPERPARAMETER

Our training setup varied slightly depending on the model type: event-based models were trained for 20 epochs, while segment-based models were trained for 10 epochs due to computational constraints. We used the Adam optimizer with a learning rate of 0.0003 and a batch size of 32. To address class imbalance, we employed weighted random sampling during training. Binary classification tasks used Binary Cross-Entropy loss, while multi-class tasks were trained with Cross-Entropy loss. Training data are processed if necessary, such as converting to time-frequency representations, before being fed into the model. Validation was performed after each epoch, and the best model checkpoint was selected based on validation macro-F1 score. Note that we do not perform extensive hyperparameter tuning; instead, we adopt a unified and simple configuration across experiments to ensure consistency and reproducibility.

### E.5    COMPUTE RESOURCES AND REPRODUCIBILITY.

All experiments were conducted on a server equipped with 4 NVIDIA RTX A6000 GPUs, each with 48 GB of memory. Training time varies by model type: event-based models typically converge within 10–30 minutes, while segment-based models require approximately 1–2 hours depending on signal duration and batch size. All training was performed using a single GPU. Preprocessing and evaluation steps were executed on the same hardware. Please refer to our released codebase for full implementation specifics, including model architectures, layer dimensions, training hyperparameters, and preprocessing pipelines.

## F    ADDITIONAL EXPERIMENTS AND ANALYSES

### F.1    ABLATION STUDY ON CROSS-DATASET GENERALIZATION

To ensure that the observed performance is not driven by dataset-specific patterns, we performed three complementary analyses: (i) leave-one-dataset-out experiments at the HFO event classification

level; (ii) leave-one-dataset-out experiments for pathological brain region identification; and (iii) a permutation test on channel-level embeddings to evaluate potential dataset-specific clustering.

**Leave-one-dataset-out HFO event classification.** We evaluate model performance in a cross-dataset generalization setting using the HFO event classification task with a leave-one-dataset-out strategy. The model is trained on all datasets except one and evaluated on the held-out dataset, with F1 score as the primary metric. The diversity and comprehensiveness of the cohort contribute to the model's generalization ability. As shown in Table 9, holding out the Open-iEEG dataset results in a performance drop relative to holding out other datasets, likely due to its larger size and greater diversity, which better support model generalization in HFO classification.

Table 9: Cross-dataset generalization via leave-one-dataset-out HFO event classification.

| Held-Out | Precision | Recall | F1 |
|----------|-----------|--------|--------|
| Open-iEEG | 0.6955 | 0.6893 | 0.6227 |
| Zurich | 0.7342 | 0.7520 | 0.7416 |
| HUP | 0.6967 | 0.7647 | 0.7219 |
| SourceSink | 0.7110 | 0.7412 | 0.7221 |

**Leave-one-dataset-out pathological brain region identification.** We additionally evaluate the pathological brain region identification task under a leave-one-dataset-out setting. Note that not all centers can be meaningfully included: the Zurich dataset does not contain SOZ annotations, and HUP includes only 13 surgical patients, making outcome evaluation statistically unstable. We therefore report two leave-one-dataset-out configurations: (i) Open-iEEG (185 patients) as the held-out dataset and (ii) Source-Sink (33 patients) as the held-out dataset. We follow the same evaluation protocol as in the main paper: all threshold-dependent metrics (Precision, Recall, F1, Specificity) use the same fixed global threshold of 0.5, and threshold-free metrics (AUC and Outcome) are evaluated identically. This ensures full consistency between the leave-one-dataset-out setting and the pooled-center benchmark. As shown in Table 10, relative performance across models remains consistent across centers; the absolute performance (especially in outcome prediction) varies due to variation in the number of seizure-free patients (Open-iEEG: 51, Source-Sink: 18), and our proposed domain-transfer model, CLAP, and TimeConv-CNN, demonstrate competitive performance under these two ablation studies.

Table 10: Cross-dataset generalization via leave-one-dataset-out pathological brain region identification.

| Held-Out | Model | Precision | Recall | F1 | Specificity | AUC | Outcome |
|----------|-------|-----------|--------|--------|-------------|--------|---------|
| | TimeConv-CNN | 0.5912 | 0.6393 | 0.5936 | 0.7383 | 0.6983 | 0.5325 |
| Open-iEEG | CLAP | 0.5785 | 0.6207 | 0.5781 | 0.7284 | 0.6560 | 0.5122 |
| | SEEGNET | 0.5746 | 0.6335 | 0.5279 | 0.5608 | 0.6650 | 0.4628 |
| | TimeConv-CNN | 0.5995 | 0.5857 | 0.5902 | 0.8392 | 0.6031 | 0.7740 |
| Source-Sink | CLAP | 0.6109 | 0.6063 | 0.6083 | 0.8168 | 0.6222 | 0.7592 |
| | SEEG-NET | 0.6965 | 0.5999 | 0.6100 | 0.9430 | 0.6331 | 0.7740 |

**Permutation test on channel-level representations.** In addition to the leave-one-dataset-out evaluations, we examine whether channel-level data distributions exhibit strong dataset-specific separation, which could confound pattern discovery. To examine this, we sample 500 one-minute segments per dataset (2000 total), generate time–frequency plots, and extract embeddings with a ViT backbone (`google/vit-base-patch16-224`). We then cluster embeddings into $k=4$ groups using K-means clustering and quantify alignment with dataset labels via homogeneity score. To assess whether any observed alignment could arise by chance, we repeat the test with 1000 random permutations of dataset labels. The observed alignment is indistinguishable from chance ($p=0.88$), indicating that there is no significant evidence of distributional differences for channel-level representations across datasets. This complements the leave-one-dataset-out analyses: while the leave-one-out analysis demonstrates generalization in predictive performance, the permutation test verifies that channel-level embeddings are not confounded by dataset distributional differences.

## F.2    ABLATION STUDY ON CHANNEL-LEVEL SEGMENT LENGTH

To justify the usage of 1-minute segments in our channel-level tasks, we evaluate the impact of segment length through an ablation study using the TimeConv-CNN-based segment classification model with 30-second, 1-minute, and 2-minutes inputs. Evaluation is restricted to test samples with at least 2 minutes of data to ensure all configurations have sufficient input length, which is a different cohort of our original cohort. Longer segments (2 min) also yield very few test windows, making resection ratio estimates unstable, so we report per-channel classification using only the channel-level labels. Segment length influences both context and stability. As shown in Table 11, very short segments (30 seconds) lack sufficient context for reliable classification, while very long segments (2 minutes) dilute fine-grained temporal features. A 1-minute segment provides a practical trade-off, capturing meaningful neurophysiological features while maintaining stable evaluation.

Table 11: Segment length ablation study (pathological channel classification).

| Model | Precision | Recall | F1 | Specificity | AUC |
|---|---|---|---|---|---|
| TimeConv-CNN (30 s) | 0.5768 | 0.7072 | 0.5442 | 0.6592 | 0.7729 |
| TimeConv-CNN (1 min) | 0.6081 | 0.7607 | 0.6096 | 0.7478 | 0.8229 |
| TimeConv-CNN (2 mins) | 0.5919 | 0.7473 | 0.5642 | 0.6684 | 0.8052 |

## F.3    ABLATION STUDY ON SAMPLING RATE

Resampling all recordings to 1000 Hz is a design choice in our benchmarking pipeline to standardize inputs across datasets while preserving spike morphology and high-frequency structure. To assess robustness to this parameter, we performed an ablation on the resampling rate, even though lower sampling rates are not clinically meaningful for intracranial EEG interpretation. As an ablation, we additionally resampled all recordings to 500 Hz and retrained the segment-based pathological region identification task. As shown in Table 12, despite the reduced resolution, the relative performance ranking across methods remains consistent with our primary results (compared with Table 5), indicating our conclusions are robust to this parameter. Note that HFO-based methods cannot be evaluated at 500 Hz, as the sampling rate violates the Nyquist limit for ripple/fast-ripple bands. Thus, only non-HFO models were included in this ablation.

Table 12: Ablation on resampling rate for pathological brain region identification (500 Hz).

| Model | Precision | Recall | F1 | Specificity | AUC | Outcome |
|---|---|---|---|---|---|---|
| TimeConv-CNN | 0.5924 | 0.7192 | 0.5884 | 0.7371 | 0.7735 | 0.7113 |
| CLAP | 0.5997 | 0.7191 | 0.6065 | 0.7740 | 0.7843 | 0.7048 |
| SEEGNET | 0.5919 | 0.7339 | 0.5744 | 0.6946 | 0.8076 | 0.6364 |

## F.4    MODEL INTERPRETABILITY ANALYSIS

Beyond raw predictive performance, model interpretability is essential in clinical applications where trust and adoption depend on understanding whether predictions are based on physiologically meaningful features. In particular, identifying which frequency bands drive model decisions helps determine whether the network captures established neurophysiological correlates of HFOs or instead relies on spurious artifacts. To this end, we conduct a SHAP analysis to examine which frequency bands most influence the TimeConv-CNN model's predictions. As shown in Figure 4, we find that activity in the 10–30 Hz range contributes most significantly to the model's spkHFO predictions. This observation aligns with clinical knowledge, as spike components are typically concentrated within this frequency band. The agreement between the model's behavior and known neurophysiological features suggests that the model is learning meaningful patterns rather than spurious correlations.

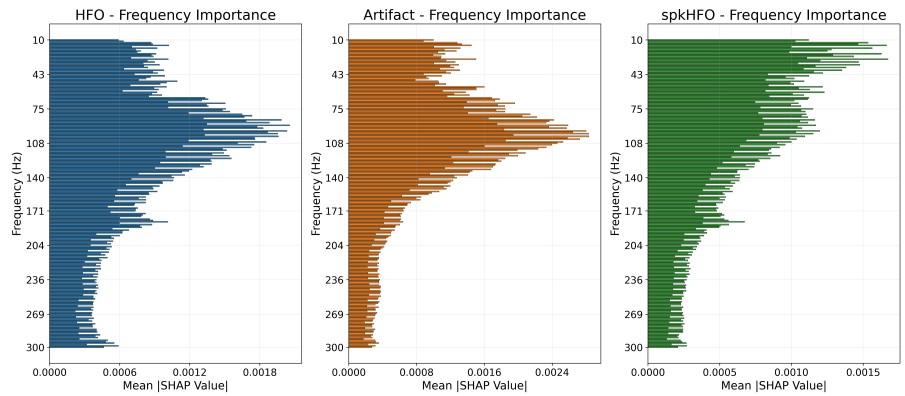

Figure 4: Model interpretation analysis using SHAP.

## F.5 ADDITIONAL BASELINES FOR PATHOLOGICAL BRAIN REGION IDENTIFICATION

To provide additional reference points beyond the main benchmarks, we include two additional baselines: an adapted EEGNet model and a linear-feature baseline. However, these baselines are not a perfect fit for iEEG settings.

**EEGNet adaptation.** EEGNet (Lawhern et al., 2018) was originally developed for scalp EEG, where recordings follow a fixed multi-channel montage and consistent spatial layout, and thus its original design does not directly transfer to iEEG. Because iEEG differs substantially in electrode configuration, spatial organization, and montage variability across patients, we adjust EEGNet to operate on a single input channel. This modification aligns with our pathological brain region identification task, where the goal is to determine whether each individual iEEG segment is pathological or not.

**Handcrafted-feature baseline.** We additionally provide a conventional machine-learning baseline built on handcrafted features extracted directly from the raw time-domain signal. These include standard time-domain descriptors (mean, median, variance, standard deviation, root-mean-square amplitude, line length, zero-crossing rate, Hjorth activity/mobility/complexity, skewness, and kurtosis) and frequency-domain features derived from a Welch-style PSD estimate (spectral entropy, peak frequency, total spectral energy, and log-bandpower across canonical bands: $\delta$ (0.5–4 Hz), $\theta$ (4–8 Hz), $\alpha$ (8–13 Hz), $\beta$ (13–30 Hz), $\gamma$ (30–80 Hz), and high-$\gamma$ (80–150 Hz)). These features have long been reported in iEEG signal analysis. We train a simple one-layer MLP with sigmoid output on these features. Table 13 shows that these baselines do not close the gap with the stronger models introduced in the main manuscript. Although the linear baseline achieves competitive outcome-prediction performance in this setting, its specificity limits practical clinical value. Overall, both EEGNet and the linear model fall short of more competitive architectures such as SEEG-NET, CLAP, and TimeConv-CNN.

Table 13: Additional baselines for pathological brain region identification (Task 2).

| Model | Precision | Recall | F1 | Specificity | AUC | Outcome |
|---|---|---|---|---|---|---|
| SEEG-NET | 0.5790 | 0.7169 | 0.5259 | 0.6049 | 0.7850 | 0.5952 |
| CLAP | 0.5936 | 0.6997 | 0.6009 | 0.7823 | 0.7684 | 0.6770 |
| TimeConv-CNN | 0.6259 | 0.7454 | 0.6469 | 0.8230 | 0.8061 | 0.7380 |
| EEGNet (adapted) | 0.5792 | 0.6957 | 0.5640 | 0.7075 | 0.7468 | 0.5942 |
| Linear baseline | 0.5323 | 0.5045 | 0.4899 | 0.9880 | 0.5045 | 0.4920 |

## G LIMITATIONS

**Subjectivity and Variability in Annotations.** While all HFO event annotations are labeled and reviewed by board-certified clinicians, inter-rater variability remains a well-recognized challenge.

For example, there is still no universally accepted definition of spkHFOs, which may introduce biases or inconsistencies in model training. Moreover, variations in labeling protocols across institutions further increase annotation variability. These sources of uncertainty underscore the need for robust, interpretable models that can operate reliably under noisy supervision.

**More Diverse Patient Cohort.** While the Omni-iEEG dataset represents one of the most comprehensive collections of invasive EEG recordings to date, curated from multiple leading epilepsy centers, it does not fully capture the global heterogeneity of epilepsy presentations and surgical practices. Important dimensions of diversity, such as broader demographic representation and varied pathology, remain underrepresented. Expanding future datasets to include a wider array of clinical contexts and patient populations will be critical for developing truly generalizable machine learning models.

**Dataset Imbalance and Biases.** Due to the inherently clinical nature of the dataset, achieving balanced representation across categories and conditions is challenging. For instance, SOZ channels are substantially outnumbered by non-SOZ channels, and certain anatomical regions or biomarker types are disproportionately represented based on clinical implantation practices. Although we introduced evaluation metrics and training strategies to mitigate these imbalances, such disparities can still bias the model toward overfitting to data-rich conditions or favoring dominant classes. These biases may limit model generalizability, particularly when deployed in settings with different clinical practices or patient populations.

**Potential Algorithmic Over-reliance.** A critical limitation lies not in the model performance itself, but in the potential misuse of machine learning systems trained on Omni-iEEG, particularly when deployed without proper human verification. Abuse of trained models in clinical workflows, such as surgical planning, could lead to inappropriate and potentially harmful decisions. Over-reliance on model outputs, especially without clinician oversight, may lead to severe consequences. Responsible use demands rigorous validation, model transparency, and active involvement of clinical experts in the decision-making process.

## H  FUTURE WORK

To address the limitations discussed above, our future work will pursue several key directions. First, we plan to expand the benchmark by incorporating additional datasets as they become publicly available or once new data collections receive IRB approval. This will enhance the diversity and generalizability of our findings. Second, we aim to evaluate a broader range of baseline methods, including unsupervised approaches that leverage graph structures and inter-channel correlations to capture latent spatial and temporal patterns in iEEG recordings.

Third, we intend to involve a wider pool of annotators from diverse academic and clinical backgrounds to label additional pathological events, using multiple biomarkers such as interictal spikes. This will enrich the dataset and improve the robustness of supervised and semi-supervised learning tasks.

Finally, while current benchmarks focus primarily on predictive performance, clinical deployment demands models that are not only accurate but also interpretable. Epilepsy surgery planning typically involves multi-disciplinary review, where transparency and trust in algorithmic outputs are critical. Even through we introduced interpretation analysis in this study, we plan to develop a more comprehensive explainable AI frameworks specifically designed for iEEG data, enabling clinicians to interpret model decisions and integrate them meaningfully into the diagnostic workflow.

## I  DETAILED DATASET SPECIFICATION AND HARMONIZATION

**Data Processing and Harmonization.** We follow the preprocessing steps recommended in prior publications (Fedele et al., 2017; Gunnarsdottir et al., 2022; Zhang et al., 2025a; Bernabei et al., 2023a) (e.g., applying bipolar montages where appropriate). All recordings are reviewed by board-certified clinicians to ensure they meet the quality standards necessary for epilepsy research. A systematic validation process checks the consistency between channel annotations and raw data, resolving misalignments or inconsistencies in channel naming conventions across datasets. Clinicians also review the complete channel lists, filtering out non-standard EEG recordings (e.g., reference, ground, EKG, or stimulation channels) and excluding those that consistently exhibit non-physiological patterns such as flat signals or excessive noise. These filtering decisions are grounded in established

clinical conventions and expert judgment—steps that are straightforward for clinicians but non-trivial for researchers without specialized training.

Beyond channel-level review, we unify and clean the metadata across sources, which vary considerably in channel annotations, surgical status, and outcome reporting. At the subject level, we include standardized attributes such as age, gender, surgical status, and postoperative outcomes, while at the channel level we provide detailed metadata including sampling frequency, modality (ECoG or SEEG), recording quality (with corrupted channels marked as `bad`), total duration, seizure onset zone (SOZ) status, resection status, and anatomical location. Together, these harmonization and quality-control steps yield a ready-to-use iEEG resource that bridges the clinical and machine learning communities.

**Contributing Datasets Overview.** For each dataset, we describe its origin, patient cohort characteristics, recording modalities, sampling frequencies, durations, and the availability of clinical annotations such as SOZ, resection zones, and surgical outcomes. We also detail the preprocessing steps undertaken to ensure consistency and data quality, as well as the strategy used for training/testing partitioning. This breakdown aims to enhance transparency and facilitate reproducibility for researchers using the Omni-iEEG dataset.

**Open iEEG Dataset.** The Open iEEG Dataset (Zhang et al., 2025a) includes iEEG recordings from two institutions: UCLA Mattel Children's Hospital (50 patients) and Children's Hospital of Michigan, Detroit (135 patients). Recordings from Detroit are exclusively ECoG, sampled at 1000 Hz, with durations ranging from around 5 to 10 minutes. Recordings from UCLA include both ECoG and SEEG, sampled at 2000 Hz, with recording durations ranging from 10 minutes up to 2 hours. Most patients have annotated seizure onset zones (SOZ), resection zones, and surgical outcomes. All recordings were acquired during the interictal sleep period. In addition, each channel is annotated with its corresponding anatomical location, enabling the dataset's use in the Anatomical Location Prediction tasks, and supporting future region-specific analyses.

**Zurich iEEG HFO Dataset.** The Zurich iEEG HFO Dataset dataset (Fedele et al., 2017) (Zurich) consists of recordings from 20 patients collected at the University Hospital Zurich. All recordings are ECoG sampled at 2000 Hz. Each patient has approximately 10–20 recordings of 5–10 minutes each. 15 patients have valid resection annotations, while none have SOZ annotations. All recordings were captured during interictal sleep stages. Resection annotations and channel status information were carefully extracted from the original publication. Following the methodology described in the original study, we also applied a bipolar montage during preprocessing.

**Epilepsy Interictal Dataset (SourceSink).** The Epilepsy Interictal Dataset dataset (Gunnarsdottir et al., 2022) comprises 39 patients collected across three institutions: National Institutes of Health (NIH), Johns Hopkins Hospital (JHH), and University of Miami Florida (UMF). This dataset includes both ECoG and SEEG recordings, with sampling rates ranging from 500 Hz to 2000 Hz. All patients have SOZ, resection zone, and surgical outcome annotations. A board-certified clinical expert reviewed the recordings and applied a bipolar montage to reduce common-mode noise. The dataset contains both sleep and awake recordings, making it suitable for tasks such as Sleep/Awake Classification. For our main benchmark, we select only recordings acquired during sleep and sampled above 1000 Hz.

**HUP Dataset.** The HUP dataset (Bernabei et al., 2023a) contains recordings from 58 patients treated at the Hospital of the University of Pennsylvania. The dataset includes both ECoG and SEEG recordings. All patients have SOZ, resection zone, and surgical outcome annotations. A board-certified clinical expert reviewed the recordings and applied a bipolar montage to reduce common-mode noise. The dataset contains both ictal and interictal recordings, making it suitable for Ictal Period Classification tasks. For our main benchmark, we include only the interictal segments with at least 1000 Hz.

Note that all datasets are collected with necessary IRB approval to ensure ethical standards. We include a comprehensive table below summarizing metadata for all the patients. For each subject, we report demographic information (age, gender), clinical annotations (SOZ, resection, surgical outcomes), inclusion in the training or testing split, and data availability indicators (presence of anatomical annotations, awake recordings, and ictal recordings). This summary offers a detailed view of the cohort composition and facilitates targeted benchmarking or subgroup analyses. Please check Table 14 for details.

Table 14: Summary of patient metadata and recording availability. Column abbreviations: SO = Surgical Outcome; SOZ = Seizure Onset Zone Annotation; Res = Resection Annotation; Anat = Anatomical Annotation; Ev = Event Annotation; Int = Interictal Recording; Ict = Ictal Recording; Awk = Awake Recording. Value abbreviations: M = Male; F = Female; Y = Yes (present); N = No (absent); - = Not applicable or not available.

| Patient Name | Dataset | Age | Gender | SO | SOZ | Res | Anat | Ev | Int | Ict | Awk | Split |
|---|---|---|---|---|---|---|---|---|---|---|---|---|
| sub-hupHUP060 | hup | 42 | F | N | Y | Y | N | N | Y | Y | N | train |
| sub-hupHUP064 | hup | 21 | M | Y | Y | Y | N | N | Y | Y | N | test |
| sub-hupHUP065 | hup | 36 | M | Y | Y | Y | N | N | Y | Y | N | train |
| sub-hupHUP070 | hup | 33 | M | Y | Y | Y | N | N | Y | Y | N | train |
| sub-hupHUP074 | hup | 25 | F | Y | Y | Y | N | N | Y | Y | N | train |
| sub-hupHUP075 | hup | 57 | F | N | Y | Y | N | N | Y | Y | N | train |
| sub-hupHUP080 | hup | 41 | F | N | Y | Y | N | N | Y | Y | N | train |
| sub-hupHUP082 | hup | 56 | F | Y | Y | Y | N | N | Y | Y | N | train |
| sub-hupHUP086 | hup | 25 | F | N | Y | Y | N | N | Y | Y | N | test |
| sub-hupHUP087 | hup | 24 | M | Y | Y | Y | N | N | Y | Y | N | test |
| sub-hupHUP088 | hup | 35 | F | Y | Y | Y | N | N | Y | Y | N | train |
| sub-hupHUP089 | hup | 29 | M | Y | Y | Y | N | N | Y | Y | N | train |
| sub-hupHUP094 | hup | 48 | F | Y | Y | Y | N | N | Y | Y | N | train |
| sub-hupHUP097 | hup | 39 | F | Y | Y | Y | N | N | Y | Y | N | train |
| sub-hupHUP105 | hup | 39 | M | Y | Y | Y | N | N | Y | Y | N | test |
| sub-hupHUP106 | hup | 45 | F | Y | Y | Y | N | N | Y | Y | N | train |
| sub-hupHUP107 | hup | 36 | M | Y | Y | Y | N | N | Y | Y | N | train |
| sub-hupHUP111 | hup | 40 | F | Y | Y | Y | N | N | Y | Y | N | train |
| sub-hupHUP112 | hup | 21 | F | N | Y | Y | N | N | Y | Y | N | train |
| sub-hupHUP114 | hup | 43 | F | N | Y | Y | N | N | Y | Y | N | train |
| sub-hupHUP116 | hup | 59 | F | Y | Y | Y | N | N | Y | Y | N | train |
| sub-hupHUP117 | hup | 39 | M | Y | Y | Y | N | N | Y | Y | N | test |
| sub-hupHUP123 | hup | 36 | M | Y | Y | Y | N | N | Y | Y | N | train |
| sub-hupHUP126 | hup | 26 | F | Y | Y | Y | N | N | Y | Y | N | train |
| sub-hupHUP130 | hup | 46 | F | Y | Y | Y | N | Y | Y | Y | N | train |
| sub-hupHUP132 | hup | 47 | F | N | N | N | N | N | Y | Y | N | test |
| sub-hupHUP133 | hup | 52 | F | N | Y | Y | N | N | Y | Y | N | train |
| sub-hupHUP134 | hup | 32 | M | Y | Y | Y | N | Y | Y | Y | N | train |
| sub-hupHUP135 | hup | 37 | M | N | Y | Y | N | N | Y | Y | N | test |
| sub-hupHUP138 | hup | 38 | M | N | Y | Y | N | N | Y | Y | N | train |
| sub-hupHUP139 | hup | 20 | M | Y | Y | Y | N | Y | Y | Y | N | train |
| sub-hupHUP140 | hup | 47 | F | Y | Y | Y | N | Y | Y | Y | N | test |
| sub-hupHUP141 | hup | 30 | M | Y | Y | Y | N | N | Y | Y | N | train |
| sub-hupHUP142 | hup | 30 | M | Y | Y | Y | N | N | Y | Y | N | train |
| sub-hupHUP144 | hup | 31 | M | Y | Y | Y | N | N | Y | Y | N | train |
| sub-hupHUP146 | hup | 16 | M | Y | Y | Y | N | Y | Y | Y | N | test |
| sub-hupHUP148 | hup | 23 | M | Y | Y | Y | N | N | Y | Y | N | train |
| sub-hupHUP150 | hup | 17 | M | Y | Y | Y | N | N | Y | Y | N | train |
| sub-hupHUP151 | hup | 33 | M | N | Y | Y | N | N | Y | Y | N | train |
| sub-hupHUP157 | hup | 25 | M | Y | Y | Y | N | N | Y | Y | N | train |
| sub-hupHUP158 | hup | 32 | M | N | Y | Y | N | N | N | Y | N | test |
| sub-hupHUP160 | hup | 45 | F | Y | Y | Y | N | N | Y | Y | N | train |
| sub-hupHUP162 | hup | 35 | F | N | Y | Y | N | N | Y | Y | N | train |
| sub-hupHUP163 | hup | 42 | F | Y | Y | Y | N | N | Y | Y | N | test |
| sub-hupHUP164 | hup | 34 | F | Y | Y | Y | N | N | Y | Y | N | test |
| sub-hupHUP165 | hup | 21 | F | N | Y | Y | N | N | Y | N | N | test |
| sub-hupHUP166 | hup | 26 | M | N | Y | Y | N | N | Y | Y | N | train |
| sub-hupHUP171 | hup | 50 | M | N | Y | Y | N | N | Y | Y | N | train |
| sub-hupHUP172 | hup | 28 | F | N | Y | Y | N | N | Y | Y | N | train |
| sub-hupHUP173 | hup | 24 | F | Y | Y | Y | N | N | Y | Y | N | train |
| sub-hupHUP177 | hup | 42 | F | Y | Y | Y | N | N | Y | Y | N | test |
| sub-hupHUP179 | hup | 20 | F | N | Y | Y | N | N | Y | Y | N | test |
| sub-hupHUP180 | hup | 28 | F | Y | Y | Y | N | N | Y | Y | N | train |
| sub-hupHUP181 | hup | 31 | F | N | Y | Y | N | N | Y | Y | N | train |
| sub-hupHUP185 | hup | 38 | M | Y | Y | Y | N | N | Y | Y | N | train |
| sub-hupHUP187 | hup | 25 | M | N | Y | Y | N | N | Y | Y | N | train |
| sub-hupHUP188 | hup | 24 | F | N | Y | Y | N | N | Y | Y | N | train |
| sub-hupHUP190 | hup | 25 | M | N | Y | Y | N | N | Y | Y | N | test |
| sub-openieegDetroit001 | openieeg | 12 | M | Y | Y | Y | Y | Y | Y | N | N | train |
| sub-openieegDetroit002 | openieeg | 8 | F | Y | Y | Y | Y | Y | Y | N | N | test |
| sub-openieegDetroit003 | openieeg | 10 | M | N | Y | Y | Y | N | Y | N | N | train |
| sub-openieegDetroit004 | openieeg | 15 | F | Y | N | Y | Y | Y | Y | N | N | test |
| sub-openieegDetroit005 | openieeg | 5 | M | Y | Y | Y | Y | Y | Y | N | N | train |
| sub-openieegDetroit006 | openieeg | 20 | M | Y | Y | Y | Y | Y | Y | N | N | train |
| sub-openieegDetroit007 | openieeg | 17 | F | Y | Y | Y | Y | N | Y | N | N | train |
| sub-openieegDetroit008 | openieeg | 6 | M | Y | Y | Y | Y | N | Y | N | N | test |
| sub-openieegDetroit009 | openieeg | 10 | F | N | N | Y | Y | N | Y | N | N | test |
| sub-openieegDetroit010 | openieeg | 11 | M | N | Y | Y | Y | N | Y | N | N | train |
| sub-openieegDetroit011 | openieeg | 17 | M | Y | Y | Y | Y | N | Y | N | N | train |
| sub-openieegDetroit012 | openieeg | 13 | F | N | Y | Y | Y | N | Y | N | N | train |
| sub-openieegDetroit013 | openieeg | 11 | F | Y | Y | Y | Y | N | Y | N | N | train |

| Patient Name | Dataset | Age | Gender | SO | SOZ | Res | Anat | Ev | Int | Ict | Awk | Split |
|---|---|---|---|---|---|---|---|---|---|---|---|---|
| sub-openieegDetroit014 | openieeg | 14 | M | Y | Y | Y | Y | N | Y | N | N | train |
| sub-openieegDetroit015 | openieeg | 5 | M | N | Y | Y | Y | N | Y | N | N | train |
| sub-openieegDetroit016 | openieeg | 8 | M | Y | Y | Y | Y | N | Y | N | N | train |
| sub-openieegDetroit017 | openieeg | 19 | F | Y | Y | Y | Y | N | Y | N | N | train |
| sub-openieegDetroit018 | openieeg | 5 | M | Y | Y | Y | Y | N | Y | N | N | train |
| sub-openieegDetroit019 | openieeg | 13 | F | N | N | Y | Y | N | Y | N | N | train |
| sub-openieegDetroit020 | openieeg | 9 | F | Y | Y | Y | Y | N | Y | N | N | train |
| sub-openieegDetroit021 | openieeg | 12 | F | Y | Y | Y | Y | N | Y | N | N | test |
| sub-openieegDetroit022 | openieeg | 10 | M | Y | Y | Y | Y | N | Y | N | N | train |
| sub-openieegDetroit023 | openieeg | 11 | F | Y | Y | Y | Y | N | Y | N | N | train |
| sub-openieegDetroit024 | openieeg | 4 | F | Y | Y | Y | Y | N | Y | N | N | test |
| sub-openieegDetroit025 | openieeg | 10 | F | Y | Y | Y | Y | N | Y | N | N | train |
| sub-openieegDetroit026 | openieeg | 16 | M | N | Y | Y | Y | N | Y | N | N | train |
| sub-openieegDetroit027 | openieeg | 15 | M | Y | Y | Y | Y | N | Y | N | N | train |
| sub-openieegDetroit028 | openieeg | 16 | M | Y | Y | Y | Y | N | Y | N | N | train |
| sub-openieegDetroit029 | openieeg | 10 | M | Y | Y | Y | Y | N | Y | N | N | test |
| sub-openieegDetroit030 | openieeg | 14 | M | Y | Y | Y | Y | N | Y | N | N | train |
| sub-openieegDetroit031 | openieeg | 7 | M | N | Y | Y | Y | N | Y | N | N | train |
| sub-openieegDetroit032 | openieeg | 28 | F | Y | Y | Y | Y | N | Y | N | N | test |
| sub-openieegDetroit033 | openieeg | 17 | M | Y | Y | Y | Y | N | Y | N | N | train |
| sub-openieegDetroit034 | openieeg | 17 | M | Y | Y | Y | Y | N | Y | N | N | test |
| sub-openieegDetroit035 | openieeg | 30 | M | Y | Y | Y | Y | N | Y | N | N | train |
| sub-openieegDetroit036 | openieeg | 10 | M | Y | Y | Y | Y | N | Y | N | N | test |
| sub-openieegDetroit037 | openieeg | 4 | M | Y | N | Y | Y | N | Y | N | N | test |
| sub-openieegDetroit038 | openieeg | 9 | F | Y | Y | Y | Y | N | Y | N | N | test |
| sub-openieegDetroit039 | openieeg | 21 | F | Y | Y | Y | Y | N | Y | N | N | test |
| sub-openieegDetroit040 | openieeg | 12 | M | Y | Y | Y | Y | N | Y | N | N | test |
| sub-openieegDetroit041 | openieeg | 28 | M | Y | N | Y | Y | N | Y | N | N | train |
| sub-openieegDetroit042 | openieeg | 11 | F | Y | Y | Y | Y | N | Y | N | N | train |
| sub-openieegDetroit043 | openieeg | 10 | M | Y | Y | Y | Y | N | Y | N | N | train |
| sub-openieegDetroit044 | openieeg | 16 | M | Y | Y | Y | Y | N | Y | N | N | test |
| sub-openieegDetroit045 | openieeg | 6 | F | Y | Y | Y | Y | N | Y | N | N | test |
| sub-openieegDetroit046 | openieeg | 19 | M | N | Y | Y | Y | N | Y | N | N | test |
| sub-openieegDetroit047 | openieeg | 12 | F | Y | Y | Y | Y | N | Y | N | N | test |
| sub-openieegDetroit048 | openieeg | 44 | M | Y | Y | Y | Y | N | Y | N | N | test |
| sub-openieegDetroit049 | openieeg | 10 | F | Y | Y | Y | Y | N | Y | N | N | train |
| sub-openieegDetroit050 | openieeg | 15 | M | Y | N | Y | Y | N | Y | N | N | train |
| sub-openieegDetroit051 | openieeg | 4 | F | Y | Y | Y | Y | N | Y | N | N | train |
| sub-openieegDetroit052 | openieeg | 10 | F | N | Y | Y | Y | N | Y | N | N | train |
| sub-openieegDetroit053 | openieeg | 12 | F | N | Y | Y | Y | N | Y | N | N | train |
| sub-openieegDetroit054 | openieeg | 8 | M | Y | Y | Y | Y | N | Y | N | N | test |
| sub-openieegDetroit055 | openieeg | 14 | M | Y | Y | Y | Y | N | Y | N | N | train |
| sub-openieegDetroit056 | openieeg | 14 | F | N | Y | Y | Y | N | Y | N | N | train |
| sub-openieegDetroit057 | openieeg | 6 | F | Y | N | Y | Y | N | Y | N | N | test |
| sub-openieegDetroit058 | openieeg | 4 | M | Y | Y | Y | Y | N | Y | N | N | train |
| sub-openieegDetroit059 | openieeg | 8 | M | Y | Y | Y | Y | N | Y | N | N | test |
| sub-openieegDetroit060 | openieeg | 14 | F | N | Y | Y | Y | N | Y | N | N | train |
| sub-openieegDetroit061 | openieeg | 37 | F | Y | N | Y | Y | N | Y | N | N | train |
| sub-openieegDetroit062 | openieeg | 19 | M | N | Y | Y | Y | N | Y | N | N | train |
| sub-openieegDetroit063 | openieeg | 11 | F | N | Y | Y | Y | N | Y | N | N | test |
| sub-openieegDetroit064 | openieeg | 14 | F | N | Y | Y | Y | N | Y | N | N | train |
| sub-openieegDetroit065 | openieeg | 17 | M | Y | Y | Y | Y | N | Y | N | N | test |
| sub-openieegDetroit066 | openieeg | 14 | M | Y | Y | Y | Y | N | Y | N | N | train |
| sub-openieegDetroit067 | openieeg | 12 | M | Y | Y | Y | Y | N | Y | N | N | train |
| sub-openieegDetroit068 | openieeg | 11 | F | Y | Y | Y | Y | N | Y | N | N | test |
| sub-openieegDetroit069 | openieeg | 27 | F | Y | Y | Y | Y | N | Y | N | N | test |
| sub-openieegDetroit070 | openieeg | 10 | M | Y | Y | Y | Y | N | Y | N | N | test |
| sub-openieegDetroit071 | openieeg | 19 | F | Y | Y | Y | Y | N | Y | N | N | train |
| sub-openieegDetroit072 | openieeg | 16 | M | N | Y | Y | Y | N | Y | N | N | train |
| sub-openieegDetroit073 | openieeg | 37 | F | Y | N | Y | Y | N | Y | N | N | train |
| sub-openieegDetroit074 | openieeg | 13 | M | Y | Y | Y | Y | N | Y | N | N | test |
| sub-openieegDetroit075 | openieeg | 14 | M | Y | N | Y | Y | N | Y | N | N | test |
| sub-openieegDetroit076 | openieeg | 8 | M | N | Y | Y | Y | N | Y | N | N | train |
| sub-openieegDetroit077 | openieeg | 15 | M | Y | Y | Y | Y | N | Y | N | N | test |
| sub-openieegDetroit078 | openieeg | 15 | F | Y | Y | Y | Y | N | Y | N | N | train |
| sub-openieegDetroit079 | openieeg | 5 | F | N | Y | Y | Y | N | Y | N | N | test |
| sub-openieegDetroit080 | openieeg | 15 | M | Y | Y | Y | Y | N | Y | N | N | test |
| sub-openieegDetroit081 | openieeg | 4 | M | Y | Y | Y | Y | N | Y | N | N | train |
| sub-openieegDetroit082 | openieeg | 11 | F | Y | Y | Y | Y | N | Y | N | N | train |
| sub-openieegDetroit083 | openieeg | 18 | F | Y | Y | Y | Y | N | Y | N | N | train |
| sub-openieegDetroit084 | openieeg | 9 | F | Y | Y | Y | Y | N | Y | N | N | test |
| sub-openieegDetroit085 | openieeg | 17 | M | Y | Y | Y | Y | N | Y | N | N | train |
| sub-openieegDetroit086 | openieeg | 7 | F | N | Y | Y | Y | N | Y | N | N | train |
| sub-openieegDetroit087 | openieeg | 14 | M | N | Y | Y | Y | N | Y | N | N | test |
| sub-openieegDetroit088 | openieeg | 13 | M | Y | Y | Y | Y | N | Y | N | N | train |
| sub-openieegDetroit089 | openieeg | 19 | F | N | Y | Y | Y | N | Y | N | N | train |
| sub-openieegDetroit090 | openieeg | 13 | F | N | Y | Y | Y | N | Y | N | N | test |
| sub-openieegDetroit091 | openieeg | 17 | M | Y | Y | Y | Y | N | Y | N | N | test |
| sub-openieegDetroit092 | openieeg | 9 | M | N | Y | Y | Y | N | Y | N | N | train |

| Patient Name | Dataset | Age | Gender | SO | SOZ | Res | Anat | Ev | Int | Ict | Awk | Split |
|---|---|---|---|---|---|---|---|---|---|---|---|---|
| sub-openieegDetroit093 | openieeg | 41 | F | Y | Y | Y | Y | N | Y | N | N | train |
| sub-openieegDetroit094 | openieeg | 6 | F | N | Y | Y | Y | N | Y | N | N | train |
| sub-openieegDetroit095 | openieeg | 12 | F | Y | Y | Y | Y | N | Y | N | N | test |
| sub-openieegDetroit096 | openieeg | 16 | F | Y | Y | Y | Y | N | Y | N | N | test |
| sub-openieegDetroit097 | openieeg | 8 | M | N | Y | Y | Y | N | Y | N | N | train |
| sub-openieegDetroit098 | openieeg | 14 | F | Y | Y | Y | Y | N | Y | N | N | train |
| sub-openieegDetroit099 | openieeg | 13 | F | N | Y | Y | Y | N | Y | N | N | train |
| sub-openieegDetroit100 | openieeg | 17 | F | Y | Y | Y | Y | N | Y | N | N | train |
| sub-openieegDetroit101 | openieeg | 5 | F | Y | Y | Y | Y | N | Y | N | N | test |
| sub-openieegDetroit102 | openieeg | 10 | M | Y | Y | Y | Y | N | Y | N | N | test |
| sub-openieegDetroit103 | openieeg | 16 | M | Y | Y | Y | Y | N | Y | N | N | test |
| sub-openieegDetroit104 | openieeg | 15 | F | Y | Y | Y | Y | N | Y | N | N | test |
| sub-openieegDetroit105 | openieeg | 7 | M | Y | Y | Y | Y | N | Y | N | N | train |
| sub-openieegDetroit106 | openieeg | 14 | F | Y | Y | Y | Y | N | Y | N | N | test |
| sub-openieegDetroit107 | openieeg | 5 | F | Y | Y | Y | Y | N | Y | N | N | train |
| sub-openieegDetroit108 | openieeg | 16 | F | N | Y | Y | Y | N | Y | N | N | train |
| sub-openieegDetroit109 | openieeg | 13 | M | N | Y | Y | Y | N | Y | N | N | test |
| sub-openieegDetroit110 | openieeg | 11 | F | Y | Y | Y | Y | N | Y | N | N | train |
| sub-openieegDetroit111 | openieeg | 10 | F | Y | Y | Y | Y | N | Y | N | N | train |
| sub-openieegDetroit112 | openieeg | 17 | M | N | Y | Y | Y | N | Y | N | N | test |
| sub-openieegDetroit113 | openieeg | 14 | F | N | N | Y | Y | N | Y | N | N | train |
| sub-openieegDetroit114 | openieeg | 8 | F | Y | Y | Y | Y | N | Y | N | N | test |
| sub-openieegDetroit115 | openieeg | 9 | F | Y | Y | Y | Y | N | Y | N | N | train |
| sub-openieegDetroit116 | openieeg | 17 | F | Y | Y | Y | Y | N | Y | N | N | train |
| sub-openieegDetroit117 | openieeg | 12 | M | N | N | Y | Y | N | Y | N | N | train |
| sub-openieegDetroit118 | openieeg | 11 | M | Y | Y | Y | Y | N | Y | N | N | test |
| sub-openieegDetroit119 | openieeg | 12 | M | N | Y | Y | Y | N | Y | N | N | train |
| sub-openieegDetroit120 | openieeg | 11 | F | Y | Y | Y | Y | N | Y | N | N | train |
| sub-openieegDetroit121 | openieeg | 23 | M | N | Y | Y | Y | N | Y | N | N | test |
| sub-openieegDetroit122 | openieeg | 5 | M | Y | Y | Y | Y | N | Y | N | N | train |
| sub-openieegDetroit123 | openieeg | 13 | M | N | Y | Y | Y | N | Y | N | N | test |
| sub-openieegDetroit124 | openieeg | 4 | F | Y | Y | Y | Y | N | Y | N | N | train |
| sub-openieegDetroit125 | openieeg | 16 | F | Y | Y | Y | Y | N | Y | N | N | train |
| sub-openieegDetroit126 | openieeg | 8 | M | N | N | Y | Y | N | Y | N | N | test |
| sub-openieegDetroit127 | openieeg | 5 | F | Y | Y | Y | Y | N | Y | N | N | test |
| sub-openieegDetroit128 | openieeg | 5 | F | N | Y | Y | Y | N | Y | N | N | train |
| sub-openieegDetroit129 | openieeg | 7 | M | N | Y | Y | Y | N | Y | N | N | test |
| sub-openieegDetroit130 | openieeg | 16 | M | N | Y | Y | Y | N | Y | N | N | test |
| sub-openieegDetroit131 | openieeg | 15 | F | Y | Y | Y | Y | N | Y | N | N | train |
| sub-openieegDetroit132 | openieeg | 8 | M | Y | Y | Y | Y | N | Y | N | N | test |
| sub-openieegDetroit133 | openieeg | 14 | F | Y | Y | Y | Y | N | Y | N | N | train |
| sub-openieegDetroit134 | openieeg | 5 | F | Y | Y | Y | Y | N | Y | N | N | train |
| sub-openieegDetroit135 | openieeg | 13 | M | N | Y | Y | Y | N | Y | N | N | train |
| sub-openieegUCLA01 | openieeg | 20 | M | Y | Y | Y | Y | Y | Y | N | N | train |
| sub-openieegUCLA02 | openieeg | 12 | M | Y | Y | Y | Y | Y | Y | N | N | train |
| sub-openieegUCLA03 | openieeg | 19 | F | Y | Y | Y | Y | Y | Y | N | N | train |
| sub-openieegUCLA04 | openieeg | 14 | F | Y | Y | Y | Y | Y | Y | N | N | test |
| sub-openieegUCLA05 | openieeg | 9 | M | Y | Y | Y | Y | Y | Y | N | N | train |
| sub-openieegUCLA06 | openieeg | 3 | F | N | Y | Y | Y | Y | Y | N | N | test |
| sub-openieegUCLA07 | openieeg | 5 | M | Y | Y | Y | Y | Y | Y | N | N | train |
| sub-openieegUCLA08 | openieeg | 19 | F | N | Y | N | Y | Y | Y | N | N | train |
| sub-openieegUCLA09 | openieeg | 13 | M | N | Y | Y | Y | Y | Y | N | N | test |
| sub-openieegUCLA10 | openieeg | 8 | F | Y | Y | Y | Y | Y | Y | N | N | test |
| sub-openieegUCLA11 | openieeg | 4 | F | Y | Y | Y | Y | Y | Y | N | N | train |
| sub-openieegUCLA12 | openieeg | 8 | F | Y | Y | Y | Y | Y | Y | N | N | test |
| sub-openieegUCLA13 | openieeg | 18 | F | N | Y | Y | Y | Y | Y | N | N | test |
| sub-openieegUCLA14 | openieeg | 15 | F | Y | Y | Y | Y | Y | Y | N | N | train |
| sub-openieegUCLA15 | openieeg | 19 | F | - | Y | N | Y | Y | Y | N | N | test |
| sub-openieegUCLA16 | openieeg | 15 | F | Y | Y | Y | Y | Y | Y | N | N | train |
| sub-openieegUCLA17 | openieeg | 6 | M | N | Y | Y | Y | Y | Y | N | N | train |
| sub-openieegUCLA18 | openieeg | 20 | M | - | Y | N | Y | Y | Y | N | N | train |
| sub-openieegUCLA19 | openieeg | 20 | M | N | Y | Y | Y | Y | Y | N | N | test |
| sub-openieegUCLA20 | openieeg | 12 | M | N | Y | Y | Y | Y | Y | N | N | test |
| sub-openieegUCLA21 | openieeg | 14 | M | - | Y | N | Y | Y | Y | N | N | train |
| sub-openieegUCLA22 | openieeg | 22 | F | N | Y | Y | Y | N | Y | N | N | train |
| sub-openieegUCLA23 | openieeg | 20 | F | - | Y | N | Y | N | Y | N | N | test |
| sub-openieegUCLA24 | openieeg | 14 | F | - | Y | N | Y | N | Y | N | N | train |
| sub-openieegUCLA25 | openieeg | 23 | F | Y | Y | Y | Y | Y | Y | N | N | test |
| sub-openieegUCLA26 | openieeg | 20 | M | Y | Y | Y | Y | N | Y | N | N | train |
| sub-openieegUCLA27 | openieeg | 6 | F | N | Y | Y | Y | N | Y | N | N | train |
| sub-openieegUCLA28 | openieeg | 17 | M | - | Y | N | Y | N | Y | N | N | train |
| sub-openieegUCLA29 | openieeg | 13 | M | - | Y | N | Y | Y | Y | N | N | train |
| sub-openieegUCLA30 | openieeg | 9 | F | Y | Y | Y | Y | N | Y | N | N | test |
| sub-openieegUCLA31 | openieeg | 13 | M | Y | Y | Y | Y | N | Y | N | N | test |
| sub-openieegUCLA32 | openieeg | 3 | F | - | Y | N | Y | Y | Y | N | N | train |
| sub-openieegUCLA33 | openieeg | 19 | M | Y | Y | Y | Y | N | Y | N | N | train |
| sub-openieegUCLA34 | openieeg | 9 | M | N | Y | Y | Y | N | Y | N | N | test |
| sub-openieegUCLA35 | openieeg | 18 | M | - | Y | N | Y | N | Y | N | N | test |
| sub-openieegUCLA36 | openieeg | 17 | F | - | Y | N | Y | N | Y | N | N | test |

| Patient Name | Dataset | Age | Gender | SO | SOZ | Res | Anat | Ev | Int | Ict | Awk | Split |
|---|---|---|---|---|---|---|---|---|---|---|---|---|
| sub-openieegUCLA37 | openieeg | 12 | F | - | Y | N | Y | Y | Y | N | N | train |
| sub-openieegUCLA38 | openieeg | 7 | M | Y | Y | Y | Y | N | Y | N | N | test |
| sub-openieegUCLA39 | openieeg | 10 | M | - | Y | N | Y | N | Y | N | N | train |
| sub-openieegUCLA40 | openieeg | 18 | F | - | Y | N | Y | N | Y | N | N | train |
| sub-openieegUCLA41 | openieeg | 16 | F | N | Y | Y | Y | N | Y | N | N | train |
| sub-openieegUCLA42 | openieeg | 25 | F | - | Y | N | Y | N | Y | N | N | train |
| sub-openieegUCLA43 | openieeg | 17 | M | Y | Y | Y | Y | N | Y | N | N | train |
| sub-openieegUCLA44 | openieeg | 21 | M | - | Y | N | Y | Y | Y | N | N | test |
| sub-openieegUCLA45 | openieeg | 25 | F | - | Y | N | Y | N | Y | N | N | train |
| sub-openieegUCLA46 | openieeg | 2 | M | - | Y | N | Y | Y | Y | N | N | train |
| sub-openieegUCLA47 | openieeg | 16 | M | - | Y | N | Y | N | Y | N | N | test |
| sub-openieegUCLA48 | openieeg | 12 | M | - | N | N | Y | N | Y | N | N | train |
| sub-openieegUCLA49 | openieeg | 28 | M | - | Y | N | Y | Y | Y | N | N | test |
| sub-openieegUCLA50 | openieeg | 2 | M | Y | Y | Y | Y | Y | Y | N | N | train |
| sub-sourcesinkjh103 | sourcesink | - | - | N | Y | Y | N | N | Y | N | Y | train |
| sub-sourcesinkjh105 | sourcesink | - | - | Y | Y | Y | N | Y | Y | N | Y | train |
| sub-sourcesinkNIH1 | sourcesink | 57 | F | Y | Y | Y | N | Y | Y | N | N | train |
| sub-sourcesinkNIH10 | sourcesink | 25 | M | N | Y | Y | N | N | Y | N | N | train |
| sub-sourcesinkNIH11 | sourcesink | 27 | M | N | Y | Y | N | N | Y | N | N | train |
| sub-sourcesinkNIH2 | sourcesink | 31 | M | Y | Y | Y | N | N | Y | N | N | test |
| sub-sourcesinkNIH3 | sourcesink | 36 | F | Y | Y | Y | N | N | Y | N | N | train |
| sub-sourcesinkNIH4 | sourcesink | 39 | M | Y | Y | Y | N | N | Y | N | N | test |
| sub-sourcesinkNIH5 | sourcesink | 41 | M | Y | Y | Y | N | N | Y | N | N | train |
| sub-sourcesinkNIH6 | sourcesink | 20 | F | N | Y | Y | N | N | Y | N | N | train |
| sub-sourcesinkNIH7 | sourcesink | 46 | M | N | Y | Y | N | N | Y | N | N | train |
| sub-sourcesinkNIH8 | sourcesink | 37 | M | N | Y | Y | N | N | Y | N | N | test |
| sub-sourcesinkNIH9 | sourcesink | 16 | F | N | Y | Y | N | N | Y | N | N | test |
| sub-sourcesinkpt1 | sourcesink | 30 | F | Y | Y | Y | N | Y | Y | N | Y | test |
| sub-sourcesinkpt2 | sourcesink | 28 | F | Y | Y | Y | N | N | Y | N | Y | test |
| sub-sourcesinkpt3 | sourcesink | 45 | M | Y | Y | Y | N | N | Y | N | Y | train |
| sub-sourcesinkPY18N002 | sourcesink | 62 | M | N | Y | Y | N | N | Y | N | N | train |
| sub-sourcesinkPY18N007 | sourcesink | 32 | F | N | Y | N | N | N | Y | N | N | test |
| sub-sourcesinkPY18N013 | sourcesink | 24 | F | Y | Y | Y | N | Y | Y | N | N | train |
| sub-sourcesinkPY18N015 | sourcesink | - | F | Y | Y | N | N | N | Y | N | N | train |
| sub-sourcesinkPY19N012 | sourcesink | 48 | M | N | Y | N | N | N | Y | N | N | test |
| sub-sourcesinkPY19N015 | sourcesink | 23 | F | N | Y | N | N | N | Y | N | N | train |
| sub-sourcesinkPY19N023 | sourcesink | 32 | M | Y | Y | N | N | N | Y | N | N | train |
| sub-sourcesinkPY19N026 | sourcesink | 35 | F | Y | Y | N | N | N | Y | N | N | train |
| sub-sourcesinkrns002 | sourcesink | 36 | F | N | Y | N | N | N | Y | N | N | test |
| sub-sourcesinkrns003 | sourcesink | 21 | M | N | Y | N | N | N | Y | N | N | train |
| sub-sourcesinkrns004 | sourcesink | 52 | M | N | Y | N | N | N | Y | N | N | test |
| sub-sourcesinkrns005 | sourcesink | 23 | M | N | Y | N | N | N | Y | N | N | test |
| sub-sourcesinkrns006 | sourcesink | 49 | M | Y | Y | N | N | N | Y | N | N | test |
| sub-sourcesinkrns009 | sourcesink | 48 | M | N | Y | N | N | N | Y | N | N | test |
| sub-sourcesinkrns011 | sourcesink | 24 | F | N | Y | N | N | N | Y | N | N | train |
| sub-sourcesinkrns013 | sourcesink | 25 | M | N | Y | N | N | N | Y | N | N | train |
| sub-sourcesinkrns014 | sourcesink | 36 | M | N | Y | N | N | N | Y | N | N | test |
| sub-sourcesinkrns015 | sourcesink | 27 | M | N | Y | N | N | N | Y | N | N | train |
| sub-sourcesinkumf001 | sourcesink | 37 | F | Y | Y | Y | N | Y | Y | N | Y | test |
| sub-sourcesinkumf002 | sourcesink | 39 | F | N | Y | Y | N | N | Y | N | Y | test |
| sub-sourcesinkumf003 | sourcesink | 43 | M | N | Y | Y | N | N | Y | N | Y | test |
| sub-sourcesinkumf004 | sourcesink | 23 | F | Y | Y | Y | N | N | Y | N | Y | test |
| sub-sourcesinkumf005 | sourcesink | 32 | F | Y | Y | Y | N | N | Y | N | Y | train |
| sub-zurich01 | zurich | - | - | Y | N | Y | N | N | Y | N | N | train |
| sub-zurich02 | zurich | 33 | M | Y | N | N | N | N | Y | N | N | train |
| sub-zurich03 | zurich | 20 | F | Y | N | Y | N | N | Y | N | N | test |
| sub-zurich04 | zurich | 20 | F | Y | N | Y | N | N | Y | N | N | train |
| sub-zurich05 | zurich | 40 | M | Y | N | N | N | N | Y | N | N | train |
| sub-zurich06 | zurich | 48 | M | Y | N | N | N | N | Y | N | N | test |
| sub-zurich07 | zurich | 25 | M | Y | N | N | N | N | Y | N | N | test |
| sub-zurich08 | zurich | 21 | F | Y | N | N | N | N | Y | N | N | train |
| sub-zurich09 | zurich | 52 | M | N | N | Y | N | N | Y | N | N | test |
| sub-zurich10 | zurich | 37 | M | Y | N | Y | N | Y | Y | N | N | test |
| sub-zurich11 | zurich | 36 | M | Y | N | Y | N | Y | Y | N | N | train |
| sub-zurich12 | zurich | 49 | M | Y | N | Y | N | Y | Y | N | N | train |
| sub-zurich13 | zurich | 17 | M | Y | N | Y | N | N | Y | N | N | test |
| sub-zurich14 | zurich | 46 | F | Y | N | Y | N | Y | Y | N | N | test |
| sub-zurich15 | zurich | 31 | F | Y | N | Y | N | Y | Y | N | N | train |
| sub-zurich16 | zurich | 17 | F | Y | N | Y | N | N | Y | N | N | train |
| sub-zurich17 | zurich | 30 | M | N | N | Y | N | N | Y | N | N | train |
| sub-zurich18 | zurich | 40 | M | N | N | Y | N | N | Y | N | N | train |
| sub-zurich19 | zurich | 38 | M | N | N | Y | N | N | Y | N | N | test |
| sub-zurich20 | zurich | 17 | M | N | N | Y | N | N | Y | N | N | train |

