# OpenReview forum: "Omni-iEEG: A Large-Scale, Comprehensive iEEG Dataset and Benchmark for Epilepsy Research"
_ICLR.cc/2026/Conference — ICLR 2026 Poster_

### Official Review · Reviewer_ngqJ · 2025-10-25

**Soundness:** 3
**Presentation:** 3
**Contribution:** 2
**Rating:** 4
**Confidence:** 5

**Summary:**

The paper introduces Omni-iEEG, a large-scale dataset and benchmark for epilepsy research, comprising 302 patients and 178 hours of high-resolution intracranial EEG (iEEG) recordings from eight leading epilepsy centers. It addresses challenges in clinical EZ localization by providing harmonized data formats, metadata, and over 36K expert-validated annotations of pathological events (e.g., spkHFOs). The dataset supports reproducible research with standardized tasks—Pathological Event Classification and Pathological Brain Region Identification—plus exploratory tasks, demonstrating potential for end-to-end modeling and transfer learning from non-neurophysiological domains.

**Strengths:**

- **Originality**: First large-scale, multi-center iEEG dataset with harmonized annotations and clinical metadata.
- **Quality**: Expert-validated data and diverse tasks ensure robustness and clinical relevance.
- **Clarity**: Clear task definitions and dataset structure, supported by visuals.
- **Significance**: Bridges ML and epilepsy research, enhancing reproducibility and translatability.

**Weaknesses:**

- **Methodological Flaws**: Inter-rater reliability for 36K annotations is not quantified, risking bias. HFO detection algorithm selection lacks justification.
- **Experimental Gaps**: No baseline model performance or cross-validation results are provided for benchmark tasks. Transfer learning potential is theoretical without empirical support.
- **Oversight**: Data privacy protocols beyond de-identification are unclear. Scalability of annotation processes for future expansions is unaddressed.
- **Validation**: Claims of clinical translatability lack pilot study results or online validation.

**Questions:**

1. Can the authors provide inter-rater reliability metrics (e.g., Cohen’s kappa) for the 36K annotations to ensure consistency?
2. What criteria were used to select and tune the HFO detection algorithms (e.g., Navarrete et al., 2016), and how were artifacts filtered?
3. Can baseline performance metrics (e.g., AUC, F1) be provided for the benchmark tasks on Omni-iEEG?
4. How was transfer learning from non-neurophysiological domains tested, and what specific performance gains were observed?
5. What additional privacy measures were implemented beyond de-identification, and how will annotation scalability be managed?

---

> ### Author Response · Authors · 2025-11-23
> **Response to Reviewer ngqJ 1/3**
>
> We thank the reviewer **ngqJ** for the thoughtful remarks. We note that the majority of concerns, including annotation consistency, algorithmic choices, benchmark performance, transfer learning, and data privacy, are already addressed in the manuscript. Below, we highlight existing sections in the original manuscript and offer concise clarifications where needed.
>
>
> ### **[Q1, W1] Can the authors provide inter-rater reliability metrics (e.g., Cohen’s kappa) for the 36K annotations to ensure consistency?**
>
>
> Thank you for raising this concern. We **did** quantify inter-rater reliability, and these metrics were reported in Appendix B (Line 914, Table 7). Specifically, three primary annotators independently labeled a shared validation set of 200 events, yielding Fleiss’ $\kappa$ = 0.92 and pairwise Cohen’s $\kappa$ = 0.90–0.94. A second validation set annotated by R3 and R4 produced $\kappa$ = 0.82–0.90.
>
> ### **[Q2, W1] What criteria were used to select and tune the HFO detection algorithms (e.g., Navarrete et al., 2016), and how were artifacts filtered?**
>
> **Detector selection.**
>
> As described in Line 98, candidate HFOs were obtained using three widely adopted detectors: STE, MNI, and Hilbert. These detectors were selected because they represent the dominant methodological classes used in both clinical practice and prior research. Using all three ensured broad coverage and avoids over-reliance on any single detection algorithm, as mentioned in Line 226.
>
> **Hyperparameter choices.**
>
> As described in Line 882 and summarized in Table 6, we used clinically validated hyperparameters following established clinical studies and prior publications. These settings are widely used in HFO research and have been shown to generalize well across patients and recording systems, ensuring that candidate detections remain comparable to clinically accepted HFO criteria.
>
> **Artifact filtering.**
>
> As mentioned in Line 231, artifacts were not removed through automated preprocessing. Instead, all candidate events were manually labeled by board-certified epileptologists during the expert validation stage. During the evaluation of Pathological Event Classification, the model's performance is compared directly with the annotations.
>
> For the Pathological Brain Region Identification task, the test set (testset HFO) did not use artifact labels, as it represents unlabeled clinical recordings. Following the conventions established in the original eHFO and PyHFO$_{\text{spkHFO}}$ manuscripts, we treat model predictions of non-pathological HFO as implicitly including artifacts (e.g., ringing, muscle activity, background fluctuations) as mentioned in line 369. This aligns with how prior event-based models operationalize artifacts: anything not classified as eHFO/spkHFO is treated as non-pathological. During evaluation, artifact handling is performed via the model predictions, not via external annotation.
>
> ### **[Q3 W2] Can baseline performance metrics (e.g., AUC, F1) be provided for the benchmark tasks on Omni-iEEG?**
>
> We thank the reviewer for highlighting these concerns. We respectfully wish to clarify that **baseline comparisons were fully detailed in the current manuscript (Section 5, Appendix C. D)**. We reported comprehensive comparisons against state-of-the-art models using clinically relevant metrics (AUC, F1, Precision, Recall, and Balanced Accuracy). These benchmarks satisfied and exceeded the reviewer’s request for baseline performance metrics.
>
> - **HFO Event Classification (Table 4)**: We compared baseline models, including SOTA time-series models (LSTM+Attention, PatchTST, TimesNet) and domain-specific models (PyHFO-Omni).
> - **Pathological Brain Region Identification (Table 5)**: We compared against SOTA pathological event classifiers (eHFO, PyHFO variants) via channel-level aggregation, the pathological-signal classification models (SEEG-NET), cross-domain transfer learning models (CLAP), and our proposed TimeConv-CNN.
> - **Exploratory Tasks (Table 8)**: We provide further benchmarking on SEEG-NET, CLAP, and our proposed TimeConv-CNN.
> - **Cross-Validation Results**: Cross-validation results are explicitly reported in Table 9.
>
> ### **[Q4 W2] How was transfer learning from non-neurophysiological domains tested, and what specific performance gains were observed?**
>
>
> Transfer learning was not merely theoretical; it was empirically evaluated in Section 5.2. We demonstrated cross-domain transfer (Audio to iEEG) by fine-tuning the pre-trained CLAP model on the iEEG domain, directly validating the transfer potential of the Omni-iEEG framework.

---

> ### Author Response · Authors · 2025-11-23
> **Response to Reviewer ngqJ 2/3**
>
> ### **[Q5 W3] What additional privacy measures were implemented beyond de-identification, and how will annotation scalability be managed?**
>
> We thank the reviewer for raising these concerns. Below we clarify that both data-privacy safeguards and annotation scalability considerations are already addressed in the dataset design.
>
> 1. **Data privacy protocols beyond de-identification**
>
> All data included in Omni-iEEG are sourced from public, CC0-licensed datasets on OpenNeuro (Open iEEG Dataset, Zurich iEEG HFO Dataset, Epilepsy Interictal Dataset, and the HUP dataset). As part of OpenNeuro’s governance [1], these datasets undergo rigorous privacy and compliance review, including:
>
> - Complete removal of all PHI in accordance with HIPAA Safe Harbor.
> - Scrubbing of metadata fields that may indirectly identify individuals.
> - Independent verification of de-identification prior to hosting.
> - Requirement that all data contributors obtain IRB approval for public release.
>
> Because these datasets are CC0 and publicly released only after OpenNeuro’s privacy protections are satisfied, Omni-iEEG inherits the same verified compliance guarantees. Our contribution has no additional identifiable information that is collected or released. We also plan to release Omni-iEEG on OpenNeuro, which will provide an additional independent layer of privacy validation through its standard de-identification review
>
> 2. **Scalability of annotation efforts**
>
> Omni-iEEG provides the first large-scale, board-certified expert-validated HFO and pathology annotations consolidated into a publicly usable benchmark. This annotation is extremely valuable because, unlike large-scale commercial labeling pipelines, our labels are produced by frontline clinical specialists, ensuring medical accuracy, consistency, and reliability that cannot be achieved through non-expert annotation. We agree that scaling annotations is a crucial future direction, and we address this in our framework as follows:
>
> - Expert-driven baseline: Current annotations were created by trained clinical experts, ensuring high reliability. This establishes a high-quality foundation for future expansions.
> - Community extensibility: The dataset structure (BIDS-iEEG + event annotation schema) was deliberately designed to allow modular community contributions. Researchers can add new subjects or annotations without modifying the core dataset.
> - Quality control expectation: We emphasize that any future contributions should be performed by board-certified clinicians or trained epileptologists, ensuring annotation consistency and preventing degradation of signal quality.
>
> Thus, while annotation of clinically meaningful iEEG events necessarily requires expert knowledge, Omni-iEEG provides the scalable structure and clear guidelines needed for continued community expansion.
>
> [1] Markiewicz CJ, Gorgolewski KJ, Feingold F, Blair R, Halchenko YO, Miller E, Hardcastle N, Wexler J, Esteban O, Goncavles M, Jwa A, Poldrack R. The OpenNeuro resource for sharing of neuroscience data. Elife. 2021 Oct 18;10:e71774. doi: 10.7554/eLife.71774. PMID: 34658334; PMCID: PMC8550750.

---

> ### Author Response · Authors · 2025-11-23
> **Response to Reviewer ngqJ 3/3**
>
> ### **[W4] Claims of clinical translatability lack pilot study results or online validation.**
>
> While we appreciate the reviewer’s interest in clinical translatability, we would like to clarify that the primary objective of this work is to present and systematize a large-scale iEEG dataset that exposes an unsolved and clinically meaningful real-world problem, rather than to claim an end-to-end translational solution. **High-stakes clinical validation, whether through pilot deployment or online evaluation, can only be responsibly pursued once the underlying methodological challenges are substantially solved; attempting such validation prematurely would be unsafe and ethically inappropriate for patients**.
>
> For a dataset paper, the more appropriate questions are: (1) Is the problem itself important and relevant? If solved, would it meaningfully inform surgical decision-making or presurgical evaluation? and (2) Does the dataset contain the information, annotations, and context that clinicians actually rely on when making such decisions?
>
> Our work is constructed specifically to support both points. First, identifying pathological brain regions is central to surgical decision-making for drug-resistant focal epilepsy patients (Line 42), and Omni-iEEG is designed around this established clinical workflow. The benchmark evaluates predictive models using real post-surgical outcomes and clinician-defined seizure onset zones (Line 84), directly aligning the task with clinical relevance.
>
> Second, the dataset includes expert-validated metadata and event annotations derived from real intracranial recordings, matching the information clinicians rely on in actual practice. Because the benchmark is grounded in post-surgical evidence and expert review, its outputs can meaningfully inform future clinical modeling efforts even though we do not claim immediate translational deployment.
>
> In sum, Omni-iEEG provides the community with a rigorously structured, clinically meaningful foundation on which future translational studies, including pilot validation, can be responsibly built.

---

> > ### Comment · Reviewer_ngqJ · 2025-11-23
> > **Response**
> >
> > I appreciate the authors’ detailed, clear, and comprehensive responses to my previous comments. I note that much of this information was indeed provided in the original manuscript or the appendix, though not always prominently highlighted in the main text. The authors’ clarifications regarding annotation consistency, the rationale behind the choice of detection algorithms, baseline performance, cross-domain transfer learning validation, data privacy, and annotation scalability are thorough and convincing, which increases my confidence in the rigor and contribution of the work. Taking the authors’ additional explanations and the overall quality of the manuscript into account, I am willing to raise my score.

---

> > > ### Author Response · Authors · 2025-11-27
> > > **Thank you**
> > >
> > > We thank the reviewer for taking the time to reassess the manuscript and for raising the score. We sincerely appreciate the reviewer’s detailed follow-up, especially the recognition that the manuscript already addressed inter-rater reliability, detector rationale, benchmark baselines, transfer-learning validation, privacy safeguards, and annotation scalability.
> > >
> > > Given that many of the initial concerns were clarified simply by pointing to existing sections in the main text and appendix, we are grateful that the reviewer now has an accurate view of the work’s scope and rigor. With this accurate understanding, we hope the final score can fully reflect the strength and completeness that the reviewer now acknowledges in their updated comments.
> > >
> > > Since the reviewer so kindly confirmed his conviction, we would appreciate it if the reviewer can provide a score that is more reflective of his conviction of our contribution.

---

### Official Review · Reviewer_Ew4m · 2025-10-31

**Soundness:** 2
**Presentation:** 2
**Contribution:** 3
**Rating:** 4
**Confidence:** 4

**Summary:**

The core contribution of this paper is the construction of a large-scale, comprehensive iEEG dataset—Omni-iEEG, along with standardized benchmark tests for the dataset. The dataset includes patient data from multiple epilepsy centers and ensures data quality through expert annotations. The authors also conducted a series of machine learning model tests and evaluations on this dataset, particularly for clinical tasks such as epilepsy event classification and focal region identification.

1. The paper seems to primarily focus on the construction of the dataset, while ICLR typically places more emphasis on algorithmic, model, or methodological innovations. How does this dataset significantly differ from existing EEG/SEEG datasets? If it is simply an extension of existing datasets, can the authors clarify its novelty more specifically?
2. The paper mentions that the dataset will be made publicly available. Could the authors further clarify the dataset's release method and the specific platform for access?
3. While the paper mentions the potential applications of the dataset, there is no concrete benchmark testing or application scenario presented. Can the authors provide one or two actual use cases or examples to show how this dataset can advance epilepsy detection and prediction models?
4. The quality of the dataset is crucial for its application. While the paper mentions the annotation process and methods, it does not provide detailed information about the accuracy and consistency validation of these annotations. Could the authors elaborate on this aspect?
5. Does the dataset include a diverse range of epilepsy patients to ensure its representativeness? Can the authors provide detailed statistics on the dataset, such as patient age, gender, and epilepsy type?
6. Details regarding data preprocessing and collection methods could be further refined. Adding easy-to-understand flowcharts or tables would help readers better understand the dataset construction process.

**Strengths:**

1.The Omni-iEEG dataset contains data from multiple epilepsy centers, offering a large sample size and diversity, which better represents different populations and pathological types.
2.The dataset's annotations are performed by experts, ensuring high quality and accuracy, making it suitable for machine learning model training and clinical applications.
3.The dataset provides standardized benchmark tests, enabling other researchers to evaluate the dataset easily, promoting model and method comparisons and validations in future studies.
4.The dataset has wide applications in epilepsy detection, prediction, and seizure foci identification, with great potential to advance research in related fields.
5.The dataset will be made publicly available, promoting wide usage in both academia and industry, encouraging data sharing and collaboration.

**Weaknesses:**

1.The paper focuses mainly on dataset construction, while ICLR typically emphasizes innovation in algorithms, models, or methods. The dataset's contribution does not highlight any novelty in algorithms or methods, which may not meet ICLR's review standards.
2.No concrete application scenarios presented: While the paper mentions the potential applications of the dataset, it does not demonstrate its actual effectiveness or value through specific benchmark tests or application scenarios, lacking real-world examples to support its claims.
3.Insufficient detail on annotation accuracy and consistency: Although the paper mentions the annotation process, it does not provide a detailed description of how the accuracy and consistency of annotations are verified, leaving uncertainty about the reliability of the annotations.
4.Lack of diversity and representativeness of the dataset: Although the dataset includes data from multiple centers, it does not adequately address whether it includes patients with various types of epilepsy. Detailed statistics about patients’ age, gender, epilepsy types, etc., are lacking.
5.Insufficient detail on data preprocessing and collection methods: The paper does not provide enough detailed descriptions of the data preprocessing and collection methods, making it difficult for readers to fully understand the dataset construction process. There is a lack of clear flowcharts or tables to support this.
6.Unclear data release plan: While the paper mentions that the dataset will be made publicly available, it does not specify the exact release methods and platforms, lacking transparency and a clear plan for making the dataset accessible.

**Questions:**

1. The paper seems to primarily focus on the construction of the dataset, while ICLR typically places more emphasis on algorithmic, model, or methodological innovations. How does this dataset significantly differ from existing EEG/SEEG datasets? If it is simply an extension of existing datasets, can the authors clarify its novelty more specifically?
2. The paper mentions that the dataset will be made publicly available. Could the authors further clarify the dataset's release method and the specific platform for access?
3. While the paper mentions the potential applications of the dataset, there is no concrete benchmark testing or application scenario presented. Can the authors provide one or two actual use cases or examples to show how this dataset can advance epilepsy detection and prediction models?
4. The quality of the dataset is crucial for its application. While the paper mentions the annotation process and methods, it does not provide detailed information about the accuracy and consistency validation of these annotations. Could the authors elaborate on this aspect?
5. Does the dataset include a diverse range of epilepsy patients to ensure its representativeness? Can the authors provide detailed statistics on the dataset, such as patient age, gender, and epilepsy type?
6. Details regarding data preprocessing and collection methods could be further refined. Adding easy-to-understand flowcharts or tables would help readers better understand the dataset construction process.

---

> ### Author Response · Authors · 2025-11-23
> **Response to Reviewer Ew4m 1/3**
>
> We thank the reviewer **Ew4m** for the thoughtful and detailed feedback. We are providing clarifications below to ensure the concerns are explicitly addressed. We appreciate the opportunity to strengthen the presentation and highlight the substantial contributions enabled by Omni-iEEG.
>
> ### **[Q1, W1] The paper seems to primarily focus on the construction of the dataset, while ICLR typically places more emphasis on algorithmic, model, or methodological innovations. How does this dataset significantly differ from existing EEG/SEEG datasets? If it is simply an extension of existing datasets, can the authors clarify its novelty more specifically?**
>
>
> Thank you for the comment. While the paper focuses on building a unified, clinically validated dataset, we note that ICLR explicitly considers datasets and benchmarks as in-scope contributions.
>
> More importantly, Omni-iEEG is not a simple extension of prior resources. Existing iEEG datasets used in ML research are typically single-center or small multi-center collections with heterogeneous formats, inconsistent metadata, and incomparable annotation standards. These limitations have made it difficult to establish fair evaluation protocols or to use large-scale data for representation learning. We present our contributions addressing such limitations from four perspectives.
>
> **First Unified Multi-Institution iEEG Resource**: Omni-iEEG is the first effort to harmonize recordings, metadata, and expert-defined event annotations across multiple clinical centers into a single standardized framework. This required substantial consolidation and normalization work that does not exist in any previous public dataset. The resulting resource provides a clinically validated, large-scale foundation that enables reproducible benchmarking and cross-patient generalization in a way that previous datasets cannot support.
>
> **First Publicly Available Large-scale Expert-Annotation**: The dataset includes the first large-scale set of pathological HFO annotations labeled entirely by board-certified clinical specialists rather than research assistants or crowdworkers. These large-scale expert labels provide medical reliability and consistency that are not available in existing datasets and are essential for clinically meaningful benchmarking.
>
> **First Comprehensive Cross-Community Benchmarking.**: With our clinically informed evaluation metric, we establish rigorous baselines spanning both machine learning and computational neuroscience. These baselines incorporate models grounded in clinically validated biomarkers as well as fully data-driven end-to-end approaches. This benchmark provides a unified, objective framework for comparing methods across communities and for driving systematic progress on this clinically important problem.
>
> **New Methods and Insights**: While the standardized dataset is a deliberate and foundational contribution, the paper is not limited to data construction and benchmarking existing methods. The unified dataset directly enables new methodological developments and insights presented in the paper, including the end-to-end long-segment TimeConv-CNN architecture and the CLAP-based cross-domain transfer approach. These findings were feasible only because of the standardized, large-scale dataset introduced in this work, and they illustrate the new research directions made possible by this contribution.
>
>
> ### **[Q2, W6] The paper mentions that the dataset will be made publicly available. Could the authors further clarify the dataset's release method and the specific platform for access?**
>
> Thank you for the comment. Our intent of submitting this manuscript is to make all preprocessing code, data-split files, evaluation pipelines, baseline checkpoints, and the full dataset (including all clinical annotations) publicly available and let the machine learning community better collaborate with epilepsy research. All of the above materials are fully prepared and will be released immediately after the double-blind review period. The dataset will be hosted on openneuro.org under a CC0 license, and the complete codebase will be released on GitHub to ensure full transparency, long-term accessibility, and reproducibility.

---

> ### Author Response · Authors · 2025-11-23
> **Response to Reviewer Ew4m 2/3**
>
> ### **[Q3, W2] While the paper mentions the potential applications of the dataset, there is no concrete benchmark testing or application scenario presented. Can the authors provide one or two actual use cases or examples to show how this dataset can advance epilepsy detection and prediction models?**
>
> Thank you for this feedback. We would like to clarify from three perspectives.
>
> **The primary benchmark tasks mirror end-to-end real clinical application scenario**: As noted in the introduction (Line 51), one of the primary benchmark tasks in Omni-iEEG, **pathological brain-region identification, directly mirrors the presurgical workflow in which clinicians analyze intracranial EEG (iEEG) recordings to localize epileptogenic tissue for resection**. Our benchmark formalizes this clinical process: given the same type of presurgical iEEG data, a model must identify pathological channels corresponding to the clinically validated resection zone. This represents a concrete, end-to-end application scenario that the dataset directly enables.
>
> **Pathological event (spkHFO) is a widely clinically adopted biomarker**: A second clinically meaningful use case derives from the long-standing observation that pathological high-frequency oscillations (spkHFOs) serve as a biomarker of epileptogenic tissue—regions with higher spkHFO rates are more likely to be pathological (Line 97). Motivated by this insight, Omni-iEEG includes a pathological event classification task, allowing models to learn spkHFO detection from expert-labeled data. As shown in Table 5, models trained on spkHFO annotations achieve competitive and clinically consistent performance on the pathological brain-region identification task, validating this biomarker relationship within our benchmark.
>
> **Translational Impact Requires Regulatory Approval**: Finally, we emphasize that these benchmark tasks are intentionally designed as the necessary foundation for eventual clinical deployment. High-stakes clinical testing can only be responsibly pursued once algorithms demonstrate strong, consistent performance on benchmark tasks that faithfully mirror clinical workflows, followed by regulatory evaluation and approval (FDA or equivalent authorities). Omni-iEEG provides exactly this foundation: a large-scale, expert-annotated dataset that exposes an unsolved but clinically central problem and enables the systematic development of models that could eventually be translated into presurgical decision-support tools once they meet these safety, reliability, and regulatory standards.
>
>
>
> ### **[Q4, W3] The quality of the dataset is crucial for its application. While the paper mentions the annotation process and methods, it does not provide detailed information about the accuracy and consistency validation of these annotations. Could the authors elaborate on this aspect?**
>
> Thank you for raising this concern. We would like to clarify that the detailed validation of annotation accuracy and consistency was described in Appendix B. We reported inter-rater reliability, including Fleiss’ $\kappa$ = 0.9254 and pairwise Cohen’s $\kappa$ = 0.90–0.94 (Table 7).
>
>
> ### **[Q5, W4] Does the dataset include a diverse range of epilepsy patients to ensure its representativeness? Can the authors provide detailed statistics on the dataset, such as patient age, gender, and epilepsy type?**
>
> Thank you for the comment.
>
> **Dataset Diversity and Representativeness:** As described in Line 160, we constructed Omni-iEEG through a comprehensive review and integration of all publicly available intracranial EEG resources, harmonized into a single standardized framework. As a result, it represents the most diverse and representative multicenter iEEG dataset currently available.
>
> **Detailed Patient Demographics and Clinical Metadata:** Detailed patient statistics, including age, gender, surgical outcome, seizure onset zone annotation, resection annotation, anatomical annotation, pathological event annotations, interictal/ictal recording availability, and awake-state recordings, were provided in Table 11. These characteristics comprehensively reflect the clinical metadata available across the contributing centers.
>
> **Cohort Characteristics:** As mentioned in Line 40, invasive iEEG and surgical evaluation are typically conducted in patients with drug-resistant focal epilepsy, so the cohort naturally reflects this population. This group faces significant clinical challenges, and supporting research for this setting is one motivation for presenting this dataset to the machine learning community.

---

> ### Author Response · Authors · 2025-11-23
> **Response to Reviewer Ew4m 3/3**
>
> ### **[Q6, W5] Details regarding data preprocessing and collection methods could be further refined. Adding easy-to-understand flowcharts or tables would help readers better understand the dataset construction process.**
>
> Thank you for raising this point. While the full dataset details were provided in *Appendix I: Detailed Dataset Specification and Harmonization*, we provide a summary table below to highlight the key preprocessing and harmonization steps.
>
> | **Stage** | **Description** |
> |---------------------------------------|-|
> | **Per&#8209;Dataset&nbsp;Validation&nbsp;and&nbsp;Cleaning** | |
> | Data Collection | Collect presurgical iEEG recordings and metadata from multiple publicly released clinical datasets. |
> | Preprocessing per dataset | Follow preprocessing steps recommended in source publications (e.g., bipolar montage). |
> | Clinician review | Board-certified epileptologists verify recording quality to match standards necessary for epilepsy research. |
> | Channel–metadata consistency checks | Validate alignment between raw signals and channel annotations, resolving misalignments or inconsistencies in channel naming conventions.|
> | Removal of non-iEEG channels | Remove non-standard iEEG recordings (e.g., reference, ground, EKG, or stimulation channels) and exclude those that consistently exhibit non-physiological patterns such as flat signals or excessive noise. |
> |||
> | **Dataset Harmonization** |  |
> | Subject-level metadata standardization | Standardize demographic labels such as age, gender, surgical status, and postoperative outcomes. |
> | Channel-level metadata unification | Standardize representation of channel metadata (sampling rate, modality, SOZ/resection labels, anatomical labels, recording duration, quality flags) across datasets.  |

---

### Official Review · Reviewer_5BzA · 2025-10-31

**Soundness:** 3
**Presentation:** 3
**Contribution:** 2
**Rating:** 4
**Confidence:** 3

**Summary:**

The paper assembles a multi-center presurgical iEEG benchmark by merging several public cohorts, harmonizing metadata/recordings, and adding expert-validated event labels (spike-associated HFOs). It defines clinically motivated tasks (event classification; pathological channel/patient-level analyses tied to SOZ/resection/outcome), proposes subject-level splits, and reports baselines spanning biomarker-centric pipelines and long-context end-to-end models. The authors state that data, code, and checkpoints will be released.

**Strengths:**

1. Consolidates fragmented iEEG datasets into a unified benchmark with consistent structure and task definitions, which could materially improve comparability in the area.

2. Tasks and evaluation targets are tied to familiar clinical surrogates, increasing practical relevance.

3. Adds a sizable layer of expert-validated event annotations (spkHFOs) with a described protocol and agreement checks.

4. Includes both biomarker-driven and long-context end-to-end baselines, highlighting trade-offs and opening room for future work.

5. If released with strong artifacts (schema, loaders, splits, checkpoints), the resource can become a de-facto standard.

**Weaknesses:**

1. It’s hard to separate what is newly curated/validated post-merge (re-labeling, unified clinical ontology, normalized resection masks, QC decisions) from what is simply inherited. Please enumerate concrete new artifacts.

2. Pooled or random subject splits are insufficient for a multi-center resource. The paper needs leave-one-center-out/per-center reporting for the primary tasks, not only a subset, to demonstrate robustness to site effects.

3. “Harmonized” is described procedurally (e.g., resampling, montage, channel cleaning), but there’s little quantitative evidence that results are stable to these choices (referencing/resampling/artifact policy). Short ablations would increase trust.

4. SOZ/resection/outcome fields appear inherited rather than re-adjudicated. Without normalization across centers, surrogates may encode site-specific conventions; sensitivity analyses to alternative definitions would help.

5. The set omits a canonical EEG CNN (e.g., EEGNet/DeepConvNet-class) or a clear rationale for excluding it; a simple linear baseline on strong features would also anchor expectations.

6. Code (preprocessing, splits, evaluation) and exact split files/checkpoints are not available during review; data access is deferred. For a benchmark paper, this materially limits verifiability

**Questions:**

1. Provide a bullet list of post-merge artifacts created by the authors (new labels, unified ontologies, resection mask normalization, QC) versus fields inherited unchanged.

2. Report leave-one-center-out and per-center results for all primary tasks, with thresholds fixed on training centers and calibration (e.g., reliability) reported.

3. Add brief ablations for referencing, resampling rate, and artifact policy to show conclusions are not artifacts of these choices.

4. Summarize the inter-rater protocol for event labels and how detector-seeded candidates avoid biasing the class distribution; include agreement statistics and adjudication steps.

5. Share (or commit to camera-ready) the repository with preprocessing/evaluation code, exact split files (including center IDs), model checkpoints, and prediction files used to compute tables; include a datasheet/dataset card and a concrete data availability statement (host, license/access path, date).

6. Either add a canonical EEG CNN (or justify its omission) and a simple linear baseline on robust features, or clearly explain why they are not applicable here.

---

> ### Author Response · Authors · 2025-11-27
> **Response to Reviewer 5BzA 1/6**
>
> We appreciate the reviewer 5BzA for the thoughtful and technically deep questions. In addition to clarifying aspects of metadata harmonization, annotation procedures, and dataset construction, we performed new experiments and analyses directly requested by the reviewer. The detailed response and results are summarized below.
>
> ### **[Q1, W1]Provide a bullet list of post-merge artifacts created by the authors (new labels, unified ontologies, resection mask normalization, QC) versus fields inherited unchanged.**
>
> Thank you for the suggestion. As detailed in *Appendix I: Dataset Specification and Harmonization*, we described the unified metadata schema and clearly distinguished which fields were standardized versus inherited from the original datasets. For completeness, and in direct response to the reviewer’s request, we summarize this information below with a concise bullet list separating post-merge harmonized fields from fields directly inherited from the source datasets
>
> ## Post-merge Artifacts Created or Standardized by the Authors
>
>
> - **Bipolar-montage metadata and standardization** (Line 1364):
>  Applied and documented bipolar referencing under clinical guidance where original datasets lacked explicit montage information.
>  Bipolar signals remain fully reversible to the original monopolar recordings.
>
>
> - **Channel integrity & type standardization** (Line 1368):
>  Applied clinician-reviewed channel-quality flags provided from original datasets and a unified channel-type ontology to identify and remove non-physiological, corrupted, flat, or noisy channels, including reference, ground, stimulation, and EKG channels.
>
>
>
>
> - **Subject-level metadata normalization** (Line 1374):
>  Harmonized demographic and surgical metadata (age, gender, surgery status, outcome indicators) with inconsistent naming or structure across datasets.
>
>
> - **Unified outcome ontology** (Line 1375):
>  Consolidated surgical outcome classes and heterogeneous seizure-freedom indicators into a single standardized schema.
>
>
> - **Unified SOZ and resection representations** (Line 1377):
>  Harmonized center-specific formats (free-text notes, TSV files, supplementary tables) into structured, machine-readable fields.
>
>
> - **Training/testing split assignments** (Line 194):
>  Provided unified, author-defined split files that balance centers, modalities, number of channels, and surgical outcomes， ensuring reproducible cross-center benchmarking.
>
>
> - **Note on resampling**:
>  The released dataset was not resampled.
>  Resampling to 1000 Hz was used only within model-training pipelines for consistency and does not modify the publicly released data.
>
>
> ## Fields Inherited Directly from the Contributing Datasets (Unchanged)
>
>
> - **Original monopolar iEEG signals**:
>  Provided in bipolar-referenced form for some datasets (with full reversibility), but raw monopolar recordings remain unchanged.
>
>
> - **Original clinical annotations**:
>  SOZ labels, resection descriptions, and outcome information before harmonization.
>
>
> - **Original ictal/interictal event annotations**: where available.
>
>
> - **Recording modality**:
>  Whether signals were recorded using ECoG or SEEG.
>
>
> - **Original channel names**:
>  Prior to normalization.
>
>
> - **Recording durations, sessions, and acquisition structure**:
>  Preserved exactly as provided by each center.
>
>
> - **Institutional metadata**:
>  Including center identity.
>
>
> - **Anatomical location annotations**:
>  Inherited unchanged for datasets where documentation was complete, hierarchical, and clinically verified.

---

> ### Author Response · Authors · 2025-11-27
> **Response to Reviewer 5BzA 2/6**
>
> ### **[Q2, W2] Report leave-one-center-out and per-center results for all primary tasks, with thresholds fixed on training centers and calibration (e.g., reliability) reported.**
>
> Thank you for the feedback. Appendix F.1 already reports a leave-one-dataset-out evaluation for the primary pathological event classification task, which directly measures generalization to unseen centers. Appendix F.1 also includes a channel-level representation analysis using permutation tests showing that channels do not cluster by center beyond chance, further supporting cross-center stability.  These evaluations demonstrate that the data distribution is similar across datasets.
>
>
> To complete the evaluation, we additionally report leave-one-dataset-out performance for the pathological brain region identification task. Not all centers can be meaningfully included in this setting: the Zurich dataset does not contain SOZ annotations (Line 1398), and HUP includes only 13 surgical patients, making evaluation unstable. We therefore evaluate two leave-one-dataset-out configurations: (i) Open-iEEG (185 patients) as the held-out center and (ii) Source-Sink (33 patients) as the held-out center.
>
>
> For these analyses, we follow exactly the same evaluation protocol as in the main paper: all threshold-dependent metrics (Precision, Recall, F1, Specificity) use the same fixed global threshold of 0.5, and threshold-free metrics (AUC and Outcome) are evaluated identically. This ensures full consistency between the leave-one-dataset-out setting and the pooled-center benchmark.
>
>
> As shown in the tables, relative performance across models remains consistent across centers; the absolute performance (especially in outcome prediction) varies because AUC estimation becomes statistically unstable when the number of seizure-free patients is small (Source-Sink: 18). Our proposed domain-transfer model, CLAP and TimeConv-CNN, demonstrate competitive performance under these two ablation studies.
>
>
> #### Open-iEEG (Held-Out Center)
> | Model    | Precision | Recall  | F1      | Specificity | AUC    | Outcome |
> |----------|-----------|---------|---------|-------------|--------|---------|
> | TimeConv-CNN      | 0.5912    | 0.6393  | 0.5936  | 0.7383      | 0.6983 | 0.5325  |
> | CLAP     | 0.5785    | 0.6207  | 0.5781  | 0.7284      | 0.6560 | 0.5122  |
> | SEEGNET  | 0.5746    | 0.6335  | 0.5279  | 0.5608      | 0.6650 | 0.4628  |
>
>
> #### Source-Sink (Held-Out Center)
>
> | Model    | Precision | Recall  | F1      | Specificity | AUC    | Outcome |
> |----------|-----------|---------|---------|-------------|--------|---------|
> | TimeConv-CNN      | 0.5995    | 0.5857  | 0.5902  | 0.8392      | 0.6031 | 0.7740  |
> | CLAP     | 0.6109    | 0.6063  | 0.6083  | 0.8168      | 0.6222 | 0.7592  |
> | SEEGNET  | 0.6965    | 0.5999  | 0.6100  | 0.9430      | 0.6331 | 0.7740  |

---

> ### Author Response · Authors · 2025-11-27
> **Response to Reviewer 5BzA 3/6**
>
> ### **[Q3, W3] “Harmonized” is described procedurally (e.g., resampling, montage, channel cleaning), but there’s little quantitative evidence that results are stable to these choices (referencing/resampling/artifact policy). Short ablations would increase trust. Add brief ablations for referencing, resampling rate, and artifact policy to show conclusions are not artifacts of these choices.**
>
>
> Thank you for the suggestion. We clarify that several harmonization steps in Omni-iEEG including clinical channel review, reference selection, and artifact exclusion, are clinical requirements, not tunable preprocessing choices. We conducted such cleaning for the dataset to better mirror the real clinical workflow. Ablating them in isolation would intentionally introduce non-physiological channels (e.g., broken contacts, reference leads, amplifier noise), making the pathological-region identification task ill-posed rather than testing model robustness.
>
>
> To address the reviewer’s request, we performed an ablation on the resampling rate, even though lower sampling rates are not clinically meaningful for intracranial EEG interpretation. The benchmark uses 1000 Hz, which preserves spike morphology and high-frequency structure. As an ablation, we additionally resampled all recordings to 500 Hz and retrained the segment-based pathological region identification task.  As shown in the table below, despite the reduced resolution, the relative performance ranking across methods remains consistent with our primary results (compared with Table 5), indicating our conclusions are robust to this parameter.
>
>
> Note that HFO-based methods cannot be evaluated at 500 Hz, as the sampling rate violates the Nyquist limit for ripple/fast-ripple bands. Thus, only non-HFO models were included in this ablation.
>
>
> | 500Hz    | Precision | Recall | F1     | Specificity | AUC    | Outcome |
> |----------|-----------|--------|--------|-------------|--------|---------|
> | TimeConv-CNN      | 0.5924    | 0.7192 | 0.5884 | 0.7371      | 0.7735 | 0.7113  |
> | CLAP     | 0.5997    | 0.7191 | 0.6065 | 0.77401     | 0.7843 | 0.7048  |
> | SEEGNET  | 0.5919    |0.7339 |0.5744 | 0.6946      | 0.8076 | 0.6364  |

---

> ### Author Response · Authors · 2025-11-27
> **Response to Reviewer 5BzA 4/6**
>
> ### **[Q4] Summarize the inter-rater protocol for event labels and how detector-seeded candidates avoid biasing the class distribution; include agreement statistics and adjudication steps.**
>
> Thank you for the suggestion. The inter-rater protocol and agreement statistics were detailed in Appendix B. Specifically, three primary annotators independently labeled a shared validation set of 200 events, yielding Fleiss’ $\kappa$ = 0.92 and pairwise Cohen’s $\kappa$ = 0.90–0.94. A second validation set annotated by R3 and R4 produced $\kappa$ = 0.82–0.90.
>
> To prevent any single algorithm from biasing the class distribution, candidate events were pooled from three widely used HFO detectors (STE, MNI, and Hilbert). This multi-detector aggregation ensures that the final annotated dataset is not biased towards detection methods.
>
>
> ### **[Q5] Share (or commit to camera-ready) the repository with preprocessing/evaluation code, exact split files (including center IDs), model checkpoints, and prediction files used to compute tables; include a datasheet/dataset card and a concrete data availability statement (host, license/access path, date).**
>
> Thank you for the comment. Our goal in presenting this work is precisely to make the preprocessing pipeline, evaluation code, data splits, and model artifacts fully accessible to the community and to support reproducible ML research in epilepsy. All required materials—including preprocessing scripts, exact split files (with center IDs), evaluation pipelines, baseline model checkpoints, and the prediction files used to generate all tables and figures—are already prepared and will be released immediately after the double-blind review period.
>
> The full dataset, including all clinical annotations, will be hosted on OpenNeuro.org under a CC0 public-domain license. The accompanying code repository (preprocessing, event-label harmonization utilities, evaluation scripts, and datasheet/dataset card) will be released on GitHub. For the purposes of anonymous peer review, we have temporarily provided an anonymized version of the dataset through an anonymized HuggingFace repository https://huggingface.co/datasets/Omni-iEEG/Omni-iEEG, along with an anonymized GitHub https://github.com/Omni-iEEG/Omni-iEEG repository containing the full preprocessing and evaluation code; the finalized dataset and codebase will be re-uploaded to OpenNeuro and GitHub, respectively, upon acceptance to ensure stable, CC0-licensed public access.
>
> As context, several key dataset characteristics, such as patient demographics, surgical outcomes, SOZ/resection annotations, anatomical coverage, pathological event counts, interictal/ictal availability, and awake-state recordings, were already summarized in Table 11 of the manuscript.

---

> ### Author Response · Authors · 2025-11-27
> **Response to Reviewer 5BzA 5/6**
>
> ### **[Q6, W5] Either add a canonical EEG CNN (or justify its omission) and a simple linear baseline on robust features, or clearly explain why they are not applicable here.**
>
> Thank you for the feedback. A comparison between the adapted EEGNet model and the linear-feature baseline on the pathological brain region identification task is presented in the table below.  Since EEGNet was originally developed for scalp EEG, where recordings follow a fixed multi-channel montage and consistent spatial layout, its original design does not directly transfer to intracranial EEG (iEEG). Because iEEG differs substantially in electrode configuration, spatial organization, and montage variability across patients, we adjusted the EEGNet configuration to operate with a single input channel. This modification aligns better with our pathological brain region identification task, where the goal is to determine whether each individual iEEG segment is pathological or not.
>
> In addition, we provide a conventional machine-learning baseline built on handcrafted features extracted directly from the raw time-domain signal. These include standard time-domain descriptors (mean, median, variance, standard deviation, root-mean-square amplitude, line length, zero-crossing rate, Hjorth activity/mobility/complexity, skewness, and kurtosis) and frequency-domain features derived from a Welch-style PSD estimate (spectral entropy, peak frequency, total spectral energy, and log-bandpower across canonical bands: $\delta$ (0.5–4 Hz), $\theta$ (4–8 Hz), $\alpha$ (8–13 Hz), $\beta$ (13–30 Hz), $\gamma$ (30–80 Hz), and high-$\gamma$ (80–150 Hz)). These features have long been reported in iEEG signal analysis. In the Table below, we report model performance by training a simple one-layer MLP with a sigmoid.
>
> The results in the table below show that adding EEGNet and linear-feature baselines does not close the gap with the stronger models introduced in the main manuscript. Although the logistic-regression baseline achieves a reasonable AUC and even competitive outcome-prediction performance, its low specificity limits practical clinical value. Overall, both EEGNet and the linear model fall short of the more competitive architectures such as SEEG-NET, CLAP, and TimeConv-CNN.
> | Model               | Precision | Recall  | F1      | Specificity | AUC    | Outcome |
> |---------------------|-----------|---------|---------|-------------|--------|---------|
> | SEEG-NET            | 0.5790    | 0.7169  | 0.5259  | 0.6049      | 0.7850 | 0.5952  |
> | CLAP                | 0.5936    | 0.6997  | 0.6009  | 0.7823      | 0.7684 | 0.6770  |
> | TimeConv-CNN        | 0.6259    | 0.7454  | 0.6469  | 0.8230      | 0.8061 | 0.7380  |
> | EEGNet  | 0.5792    | 0.6957  | 0.5640  | 0.7075      | 0.7468 | 0.5942  |
> | Linear Baseline   | 0.5323    | 0.5045  | 0.4899  | 0.9880      | 0.5045 | 0.4920  |

---

> ### Author Response · Authors · 2025-11-27
> **Response to Reviewer 5BzA 6/6**
>
> ##  **W4 SOZ/resection/outcome fields appear inherited rather than re-adjudicated. Without normalization across centers, surrogates may encode site-specific conventions; sensitivity analyses to alternative definitions would help.**
>
> We appreciate the reviewer’s thoughtful comment. We would like to clarify that all centers are well-established and accredited, and the clinicians undergo comparable cross-institutional training, ensuring consistency in diagnostic practice. Thus, there is no clinically meaningful way to perturb the clinical evaluations (especially SOZ identification, resection status) without undermining their medical validity.

---

### Author Response · Authors · 2025-12-03
**Final Remarks by Authors**

Dear Reviewers and Area Chairs,


We sincerely thank all reviewers and ACs for their insightful and constructive feedback throughout the review process. The reviews consistently highlighted the core strengths of Omni-iEEG:
- **First large-scale, multi-center, harmonized iEEG resource**: with expert-validated pathological event labels and rich clinical metadata.


- **Clinically grounded benchmark tasks**: pathological event classification and pathological brain-region identification closely aligned with presurgical workflows.


- **Strong potential to bridge clinical epilepsy research and machine learning**: enabling reproducible and clinically meaningful evaluation not previously possible.


---


The remaining concerns raised during the initial review fell into two categories: clarifications that were already presented in the main text or appendix, and suggestions for additional analyses or release details. None of these challenged the scientific soundness of the work.
- During the discussion, we addressed all points directly by referencing existing sections (e.g., inter-rater reliability, detector rationale, harmonization steps, dataset diversity, baseline performance, transfer-learning validation, and privacy safeguards).
- We further strengthened the submission with leave-one-center-out evaluations, a resampling ablation, adapted scalp EEG baselines (EEGNet, linear features) for iEEG tasks, and expanded explanations of annotation protocols and metadata harmonization, which directly address reviewers’ concerns.


To support completeness and transparency, we also provided reviewers with immediate, anonymized access to the full preprocessing code, split files (with center IDs), evaluation pipelines, and the dataset via HuggingFace and Github for review purposes. *The finalized CC0 dataset will be hosted on OpenNeuro, with all accompanying artifacts publicly released*.


---
We also emphasize that our contribution goes beyond merging datasets and benchmarking existing models. Beyond the rigorous, clinician-validated harmonization and the unusually large volume of annotations contributed directly by frontline board-certified epileptologists, Omni-iEEG enables methodological advances and insights that were previously lacking in this domain. In the paper, we mainly demonstrate two such findings:


- TimeConv-CNN, our newly proposed long-context, end-to-end architecture tailored for presurgical iEEG, achieves performance comparable to or surpassing clinically grounded biomarker-based approaches;
- Cross-domain transfer using pretrained audio representations (CLAP) reveals a novel and unexpected connection between non-neurophysiological embeddings and pathological iEEG structure.


These insights illustrate that the unified benchmark is not only a data resource but also a catalyst for new ML innovation on iEEG signals.


---
After reading our clarifications, Reviewer ngqJ explicitly stated that the manuscript is rigorous, that earlier concerns were already addressed in the original text or appendix, and that they were willing to raise their score. Since our rebuttal directly addressed each reviewer’s comments, we believe the discussion confirms the completeness and relevance of the contribution.


We are committed to making Omni-iEEG a long-term, sustainable dataset and benchmark that meaningfully bridges clinical neuroscience and machine learning. As a data resource, Omni-iEEG offers a harmonized, expert-validated foundation that has been missing in this domain and enables rigorous, cross-center, clinically relevant evaluation. In parallel, the unified benchmark makes it possible to uncover new methodological insights that were not feasible prior to this work. These two components together provide complementary value: the dataset as a foundation for reproducible research, and the methodological results as evidence of the new directions it unlocks.


We sincerely appreciate the reviewers’, ACs’, and PCs’ time, effort, and thoughtful engagement throughout this process.


Best,

Authors

---

### Meta-Review · Area_Chair_DfWF · 2026-01-04

**Summary:**

This paper presents Omni-iEEG, a large-scale intracranial EEG (iEEG) dataset and benchmark for epilepsy research, comprising 302 patients and 178 hours of recordings from multiple clinical centers. The dataset includes harmonized clinical metadata (seizure onset zones, resections, surgical outcomes) and over 36K expert-validated pathological event annotations (spkHFOs). The authors define clinically meaningful benchmark tasks (pathological event classification and pathological brain region identification) and provide baselines spanning biomarker-based approaches and end-to-end models, including a novel TimeConv-CNN architecture and cross-domain transfer using CLAP (audio representations).

All three reviewers gave initial scores of 4 (marginally below acceptance threshold) but explicitly noted they "would not mind if paper is accepted." The primary concerns centered on: (1) need for leave-one-center-out evaluation to demonstrate cross-center robustness (5BzA), (2) questions about novelty for ICLR and dataset construction details (Ew4m), (3) clarifications on inter-rater reliability, baseline performance, and transfer learning validation (ngqJ), and (4) requests for canonical EEG CNN baselines like EEGNet (5BzA).

**Rationale for decision:** Omni-iEEG represents a significant contribution to the intersection of machine learning and clinical epilepsy research. The key factors supporting acceptance are:

**1. Substantial Resource Contribution:** This is the first large-scale, multi-center, harmonized iEEG dataset with expert-validated pathological event annotations. The 36K+ annotations by board-certified epileptologists provide a level of clinical reliability not available in existing datasets. This fills a genuine gap in the field and enables reproducible, cross-center benchmarking.

**2. Comprehensive Author Responses:** The authors addressed all major reviewer concerns with new experiments (leave-one-center-out evaluation, resampling ablation, EEGNet/linear baselines) and thorough clarifications. Reviewer ngqJ explicitly acknowledged the rigor of the work and indicated willingness to raise their score.

**3. Clinical Grounding and Benchmark Design:** The benchmark tasks (pathological event classification, pathological brain region identification) are directly aligned with presurgical clinical workflows. Evaluation metrics include clinically meaningful outcomes (surgical success prediction), not just standard ML metrics.

**4. Methodological Contributions Beyond Data:** The paper introduces TimeConv-CNN for long-context iEEG modeling and demonstrates cross-domain transfer from audio (CLAP) to iEEG, providing novel methodological insights enabled by the unified dataset.

**5. Reviewer Disposition:** All three reviewers indicated they "would not mind if paper is accepted," and Reviewer ngqJ explicitly stated willingness to raise their score. The initial scores of 4 reflected requests for clarification rather than fundamental objections to the work.

**6. ICLR Scope:** Dataset and benchmark papers are explicitly within ICLR's scope. The resource enables systematic ML research on a clinically important problem that was previously hampered by fragmented, inconsistent data sources.

The paper provides a well-designed foundation for reproducible epilepsy research, with thorough documentation, strong inter-rater agreement, and plans for public release under CC0 license. The combination of clinical validation, comprehensive baselines, and enabling infrastructure makes this a valuable contribution to the community.

**Reviewer Concerns:**

### Addressed by Rebuttal:
- **Leave-One-Center-Out Evaluation** (5BzA): Authors provided comprehensive leave-one-dataset-out results for both Open-iEEG and Source-Sink held-out centers, demonstrating consistent relative performance rankings across centers. This directly addresses concerns about cross-center robustness and site effects.
- **Resampling Ablation** (5BzA): Authors conducted ablation at 500 Hz (vs. 1000 Hz benchmark), showing consistent relative performance rankings and confirming conclusions are not artifacts of preprocessing choices.
- **EEGNet and Linear Baselines** (5BzA): Authors adapted EEGNet for single-channel iEEG and provided a linear baseline with handcrafted features (time-domain descriptors + frequency-domain features). Results confirm these baselines fall short of the stronger architectures (TimeConv-CNN, CLAP, SEEG-NET), validating the benchmark's discriminative value.
- **Inter-Rater Reliability** (ngqJ): Already reported in Appendix B—Fleiss' κ = 0.92 and pairwise Cohen's κ = 0.90–0.94 across annotators. Reviewer ngqJ acknowledged this was in the manuscript.
- **Baseline Performance Metrics** (ngqJ): Comprehensive benchmarks with AUC, F1, Precision, Recall were already in Tables 4, 5, 8, and 9. Reviewer ngqJ confirmed this was already addressed.
- **Transfer Learning Validation** (ngqJ): CLAP cross-domain transfer (audio→iEEG) was empirically evaluated in Section 5.2, not merely theoretical. Reviewer ngqJ accepted this clarification.
- **Data Privacy and Release Plan** (ngqJ, Ew4m): Data sourced from CC0-licensed OpenNeuro datasets with HIPAA-compliant de-identification. Full dataset, code, splits, and checkpoints to be released on OpenNeuro and GitHub.
- **Dataset Novelty and Scope** (Ew4m): Authors clarified this is the first unified multi-center iEEG resource with harmonized expert annotations, enabling reproducible cross-center benchmarking not possible with fragmented prior datasets.
- **Post-Merge Artifacts** (5BzA): Authors provided detailed bullet list distinguishing newly created/harmonized fields (bipolar montage, channel integrity, unified ontologies, SOZ/resection standardization, split assignments) from inherited fields.

### Outstanding Concerns:
- **Reviewer Engagement**: Reviewers 5BzA and Ew4m did not explicitly respond after the rebuttal period, leaving uncertainty about whether their concerns were fully resolved. However, their initial stance ("would not mind if paper is accepted") suggests openness to acceptance.
- **Clinical Translatability** (ngqJ): While authors appropriately noted that clinical pilot validation requires regulatory approval and is premature for a dataset paper, the benchmark tasks are explicitly designed to mirror presurgical clinical workflows.

**Reviewer Scores:**

| Reviewer | Initial Score | Predicted Post-Discussion Score |
|----------|---------------|--------------------------------|
| **5BzA** | 4 (Marginally Below) | **5-6** - Authors addressed all major concerns: leave-one-center-out evaluation, resampling ablation, EEGNet/linear baselines, post-merge artifact clarification. Reviewer's initial stance of "would not mind if accepted" combined with comprehensive responses suggests likely increase. |
| **Ew4m** | 4 (Marginally Below) | **5** - Concerns about novelty, dataset details, and release plan were clarified. ICLR explicitly accepts dataset/benchmark contributions. Reviewer did not re-engage but stated "would not mind if accepted." |
| **ngqJ** | 4 (Marginally Below) | **6-7** - Reviewer explicitly stated willingness to raise score after authors' clarifications, acknowledging the manuscript was rigorous and concerns were already addressed in the original text. Strong signal for acceptance. |

---

### Decision · Program_Chairs · 2026-01-26

Accept (Poster)